# Multi-state recognition pathway of the intrinsically disordered protein kinase inhibitor by protein kinase A

Cristina Olivieri[1†], Yingjie Wang[2,3†], Geoffrey C Li[2†‡], Manu V S[1], Jonggul Kim[2§], Benjamin R Stultz[4], Matthew Neibergall[4], Fernando Porcelli[5], Joseph M Muretta[1], David DT Thomas[1], Jiali Gao[2,6], Donald K Blumenthal[7], Susan S Taylor[8], Gianluigi Veglia[1,2]*

[1]Department of Biochemistry, Molecular Biology, and Biophysics, University of Minnesota, Minneapolis, United States; [2]Department of Chemistry and Supercomputing Institute, University of Minnesota, Minneapolis, United States; [3]Shenzhen Bay Laboratory, Shenzhen, China; [4]Department of Chemistry, Bethel University, Saint Paul, United States; [5]DIBAF, University of Tuscia, Largo dell' Università, Viterbo, Italy; [6]Laboratory of Computational Chemistry and Drug Design, Peking University Shenzhen Graduate School, Shenzhen, China; [7]Department of Pharmacology and Toxicology, University of Utah, Salt Lake City, United States; [8]Department of Chemistry and Biochemistry and Pharmacology, University of California, San Diego, La Jolla, United States

*For correspondence:
vegli001@umn.edu

[†]These authors contributed equally to this work

Present address: [‡]Department of Biochemistry, Vanderbilt University School of Medicine, Nashville, United States; [§]Department of Biophysics, Howard Hughes Medical Institute, University of Texas Southwestern Medical Center, Dallas, United States

Competing interests: The authors declare that no competing interests exist.

**Abstract** In the nucleus, the spatiotemporal regulation of the catalytic subunit of cAMP-dependent protein kinase A (PKA-C) is orchestrated by an intrinsically disordered protein kinase inhibitor, PKI, which recruits the CRM1/RanGTP nuclear exporting complex. How the PKA-C/PKI complex assembles and recognizes CRM1/RanGTP is not well understood. Using NMR, SAXS, fluorescence, metadynamics, and Markov model analysis, we determined the multi-state recognition pathway for PKI. After a fast binding step in which PKA-C selects PKI's most competent conformations, PKI folds upon binding through a slow conformational rearrangement within the enzyme's binding pocket. The high-affinity and pseudo-substrate regions of PKI become more structured and the transient interactions with the kinase augment the helical content of the nuclear export sequence, which is then poised to recruit the CRM1/RanGTP complex for nuclear translocation. The multistate binding mechanism featured by PKA-C/PKI complex represents a paradigm on how disordered, ancillary proteins (or protein domains) are able to operate multiple functions such as inhibiting the kinase while recruiting other regulatory proteins for nuclear export.

## Introduction

The cAMP-dependent protein kinase A (PKA) is a ubiquitous phosphoryl transferase that regulates numerous cellular signaling pathways (*Taylor et al., 2012*). As for other eukaryotic protein kinases, PKA regulatome comprises globular domains as well as extensive disordered regions, which are essential for protein-protein interactions and signaling (*Akimoto et al., 2013*; *Gógl et al., 2019*; *Pellicena and Kuriyan, 2006*). In fact, the activity and localization of PKA-C within different cellular compartments are finely regulated by ancillary proteins such as the regulatory subunits (R), A-kinase-anchoring proteins (AKAPs) (*Bauman and Scott, 2002*; *Johnson and Lewis, 2001*), and the heat-stable protein kinase A inhibitor (PKI) (*Dalton and Dewey, 2006*). While R-subunits are primarily responsible for the regulation and localization of PKA-C in the cytoplasm through interaction with

AKAPs (*Taylor et al., 2012*), PKI plays a key role in transcriptional regulation and the export of the enzyme from the nucleus to the cytoplasm (*Fantozzi et al., 1992*; *Wiley et al., 1999*).

PKI is expressed in different isoforms of 70–75 amino acids in length, which were first identified in skeletal muscle extracts (*Dalton and Dewey, 2006*). PKI isoforms contain several functional motifs: a high affinity region (HAR, residues 1–14), a pseudo-substrate recognition sequence (PSS, residues 15–22), a nuclear export signal (NES, residues 37–46), and a disordered C-terminal tail (residues 47–75) (*Figure 1A*; *Johnson et al., 2001*). The HAR is an accessory domain that increases the affinity of PKI for the enzyme, the PSS mimics the consensus sequence for the enzyme; while the NES motif mediates the recruitment by the CRM1/RanGTP complex for PKA-C translocation from the nucleus to the cytoplasm (*Figure 1D*; *Wen et al., 1995*). PKI binds PKA-C with nanomolar affinity (*Whitehouse and Walsh, 1982*), which is matched only by that of the R-subunits with whom it shares the same recognition/inhibitory sequence (*Figure 1C*; *Hofmann, 1980*). Unlike the R-subunits, PKI lacks any cAMP binding sites and its regulation of PKA-C is independent of the cyclic nucleotide-mediated signaling (*Dalton and Dewey, 2006*).

In humans, three different genes encode the corresponding PKI isoforms, PKIα, PKIβ, PKIγ (*Dalton and Dewey, 2006*). Among those, the alpha isoform (PKIα) has the highest level of

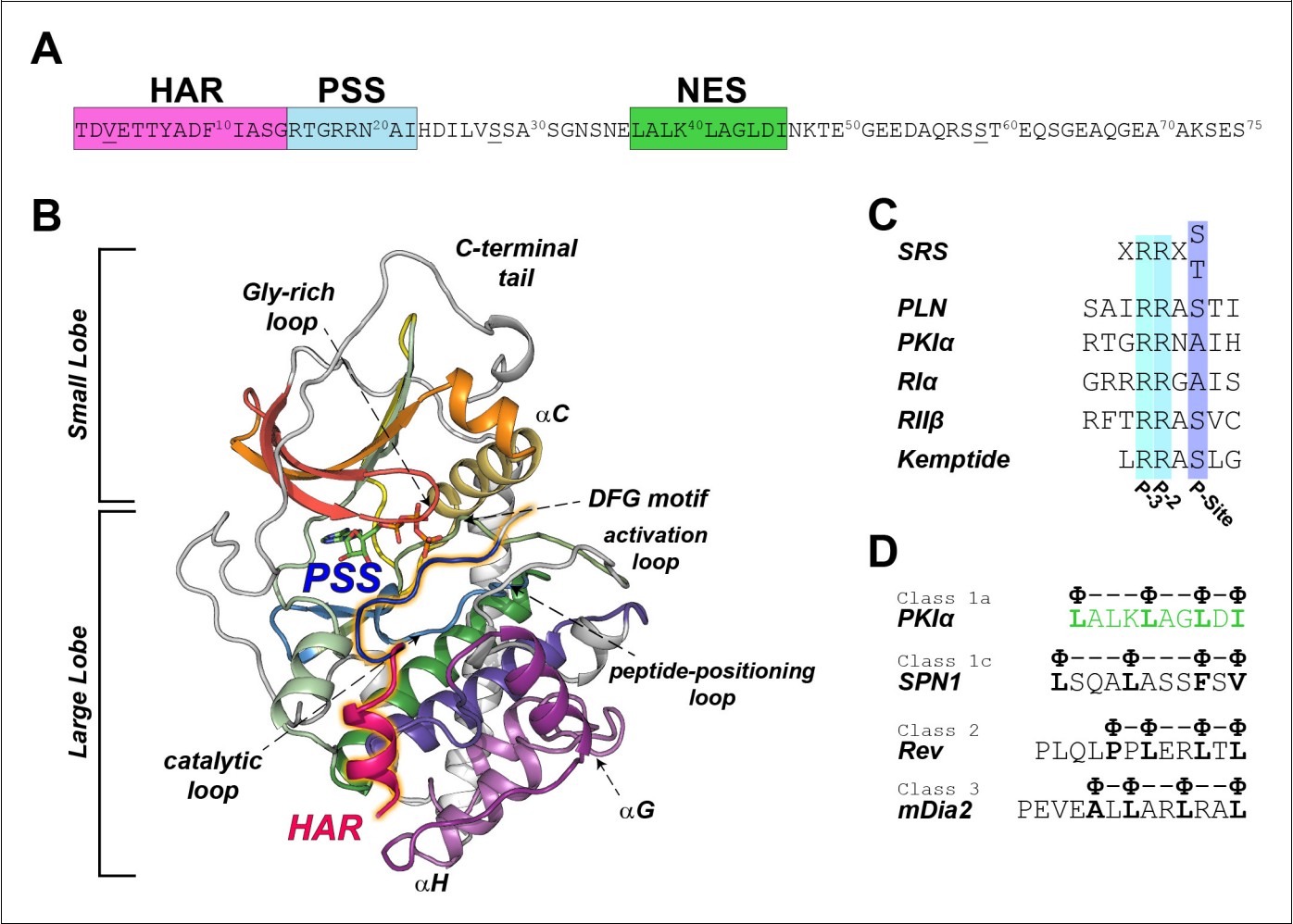

**Figure 1.** PKIα primary sequence, homology, and complex with PKA-C. (A) Primary sequence of PKIα, featuring the high affinity region (HAR), the pseudo-substrate region (PSS), and the nuclear export signal (NES). (B) X-ray structure of the PKA-C/ATP/PKI$_{5-24}$ ternary complex featuring only part of the HAR sequence and the PSS motif (PDB ID: 1ATP; *Knighton et al., 1991a*). PKA-C secondary structure elements are color-coded using Hanks and Hunter definition (*Hanks et al., 1988*). (C) Comparison of the substrate recognition sequences (SRS) for different PKA-C substrates, including PKIα, Kemptide, phospholamban (PLN), and the regulatory subunit RIα and RIIβ. D. NES sequences from different proteins (*Fung et al., 2015*; *Fung et al., 2017*).

expression in various tissues and the highest inhibitory potency [$K_i$ ~0.22 nM, (*Collins and Uhler, 1997*; *Gamm and Uhler, 1995*). The initial crystal structures of PKA-C were obtained with truncated variants of PKIα, encompassing either residues 5–22 or 5–24 with part of the HAR and PSS sequences, lacking the NES as well as the C-terminal regions. (*Figure 1B*; *Knighton et al., 1991a*; *Knighton et al., 1991b*). More importantly, there are no reports on how PKI is recognized by the kinase and its binding mechanism.

Using a combination of NMR spectroscopy, small angle X-ray scattering (SAXS), stopped-flow fluorescence, replica-averaged metadynamics (RAM), and Markov model analysis, here we provide a comprehensive view of the molecular mechanism for recognition of PKIα by PKA-C. We found that free PKIα is intrinsically disordered, with incipient α-helical segments spanning the HAR and NES regions. PKA-C recognizes and binds extended PKIα conformations, rigidifying the PSS region and increasing the helicity of the HAR and NES motifs. Transient binding kinetics show that the recognition occurs via a multi-state pathway, with an initial fast binding step followed by a slow phase in which PKIα undergoes a conformational rearrangement within the binding site. Although the C-terminal tail of PKIα interacts transiently with the kinase's C-lobe, it remains essentially disordered facilitating the recruitment by the CRM1/RanGTP complex for nuclear export. We propose that the multistate pathway for PKIα recognition by PKA-C is preparatory for the recruitment by CRM1/RanGTP for nuclear export and regulation of gene expression.

## Results

### PKIα is an intrinsically disordered protein with transient secondary and short-lived tertiary interactions

In agreement with Hauer et al. (*Hauer et al., 1999a*; *Thomas et al., 1991*), we found that the dichroic profile, the NMR fingerprint, and the dynamic NMR parameters display the typical signature of mixed α-helix and random-coil structure for PKIα (*Figure 2*, *Figure 2—figure supplement 1A*). Addition of urea to PKIα only slightly changes the chemical shift of the amides, confirming the disordered nature of the protein and suggesting the presence of only transient secondary structures (*Figure 2—figure supplement 1B*; *Felitsky et al., 2008*; *Marsh et al., 2007*). We also probed the presence of long-range contacts using $^1H_N$ paramagnetic relaxation enhancement ($^1H_N$ PRE-$\Gamma_2$) experiments (*Clore and Iwahara, 2009*; *Figure 2E–G*). For these experiments, we conjugated a nitroxide spin label (MTSL) on engineered cysteine residues at three different positions: PKIα$^{V3C}$, PKIα$^{S28C}$, and PKIα$^{S59C}$ in analogy to earlier fluorescence studies (*Hauer et al., 1999b*). We then measured the peak intensities in the presence of paramagnetic ($I_{para}$) and diamagnetic ($I_{dia}$) labels to estimate the $\Gamma_2$ relaxation rates (*Clore and Iwahara, 2009*). When the spin label was engineered at the N-terminus (PKIα$^{V3C}$), we observed a reduction of the resonance intensities near the N-terminal region and gradual effects that culminate in the middle of the linker between the PSS and NES motifs and decrease toward the C-terminus (PRE values less than two standard deviations, 2σ). The C-terminal residues remained essentially unperturbed (*Figure 2E*). On the other hand, when the spin label was positioned between the PSS and NES motifs (PKIα$^{S28C}$), we detected strong PREs near the spin label position (PRE greater than 3σ), at the N-terminus, and within the NES motif (*Figure 2F*). Finally, for the spin label at position 59, we found that approximately 75% of the residues display a reduced signal intensity, with marked effects in the PSS, linker region, NES, and C-terminus. In this case, we observed only marginal effects for the N-terminal residues (*Figure 2G*). Altogether, the CD and NMR data reveal that in absence of a binding partner PKIα samples transient secondary structures encompassing the HAR and NES motifs with short-lived tertiary conformations.

### PKA-C binding rigidifies the PSS motif and stabilizes HAR and NES helices of PKIα

To map the fingerprints of both PKA-C and PKIα in the PKA-C/ATPγN/PKIα complex, we used our previously developed method in which we combine asymmetric isotopic labeling for each binding partner with a spin-echo filter for $^{13}$C-linked amide protons in the heteronuclear single quantum coherence experiment (HSQC), [$^1$H,$^{15}$N]-carbonyl carbon label selective-HSQC or [$^1$H, $^{15}$N]-CCLS-HSQC (*Tonelli et al., 2007*; *Larsen et al., 2018*; *Figure 3—figure supplement 1*). This method makes it possible to monitor the fingerprints of two interacting proteins simultaneously using only

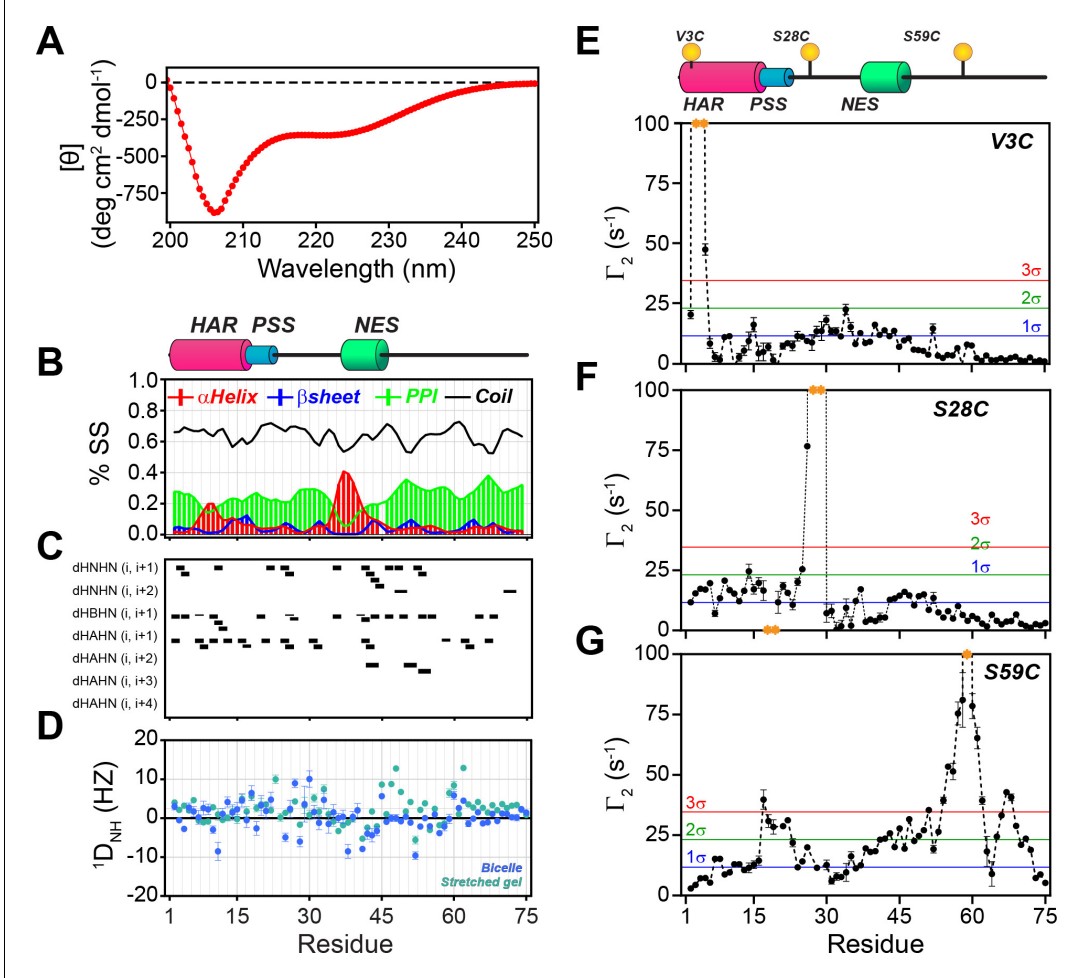

**Figure 2.** PKIα is an intrinsically disordered protein with transient tertiary contacts. (A) Far-UV circular dichroism (CD) spectrum of PKIα acquired at 25°C in native condition showing the typical CD profile for IDPs, with a minimum around 208 nm (*Wicky et al., 2017*; *Chemes et al., 2012*; *Silvers et al., 2012*). (B) Secondary structure content for free PKIα as determined from NMR chemical shifts using δ2D program (*Camilloni et al., 2012*). Note that the program neglects to calculate values for the first and last residues. (C) Plots of the [$^1$H,$^1$H] NOEs derived from the analysis of [$^1$H,$^{15}$N] NOESY-HSQC experiment. (E) Residual dipolar couplings (RDCs) for PKIα aligned in stretched polyacrylamide gels and 5% bicelles (q = 3.5). The oscillations around zero of the $^1$D$_{NH}$ values confirm the absence of a defined structure. (E-F) Intra-molecular $^1$H- $\Gamma_2$-PRE measurements of the three cysteine mutants of PKIα (*Iwahara et al., 2007*). For these experiments, a delay of 10 ms was used to calculate the intra-molecular $^1$H- $\Gamma_2$-PRE effects. The yellow spheres on the schematic of PKIα (B panel) indicate the location of the cysteine mutations. The yellow dots in each graph indicate those residues broadened beyond detection. The horizontal lines indicate one (blue), two (green), and three (red) standard deviations (σ) from the mean PRE.

The online version of this article includes the following figure supplement(s) for figure 2:

**Figure supplement 1.** PKIα is an intrinsically disordered protein.

---

one sample preparation. Upon binding to PKA-C, most of the PKIα resonances in the [$^1$H,$^{15}$N]-HQSC spectrum show only minimal changes (*Figure 3—figure supplement 1C*).

However, the HAR and PSS resonances display significant chemical shift changes (*Figure 3A,B,E*), suggesting that these regions of PKI undergo drastic structural rearrangements upon binding the kinase. This finding is also supported by the δD2 analysis (*Camilloni et al., 2012*) of the Cα and Cβ chemical shifts showing an increase in helical content of the HAR and rigidification of the PSS motif; while the rest of the peptide remain unaffected by the interaction with PKA-C (*Figure 3B*, *Figure 3—figure supplement 2A*). The analysis of the chemical shift difference between the ATPγN-bound form and the ternary complex mapped on PKA-C are reported in *Figure 3* (panel C-D). As previously reported (*Masterson et al., 2010*; *Masterson et al., 2008*), the amide fingerprint of the kinase changes significantly upon nucleotide binding (*Figure 3C*). Additional changes are observed upon

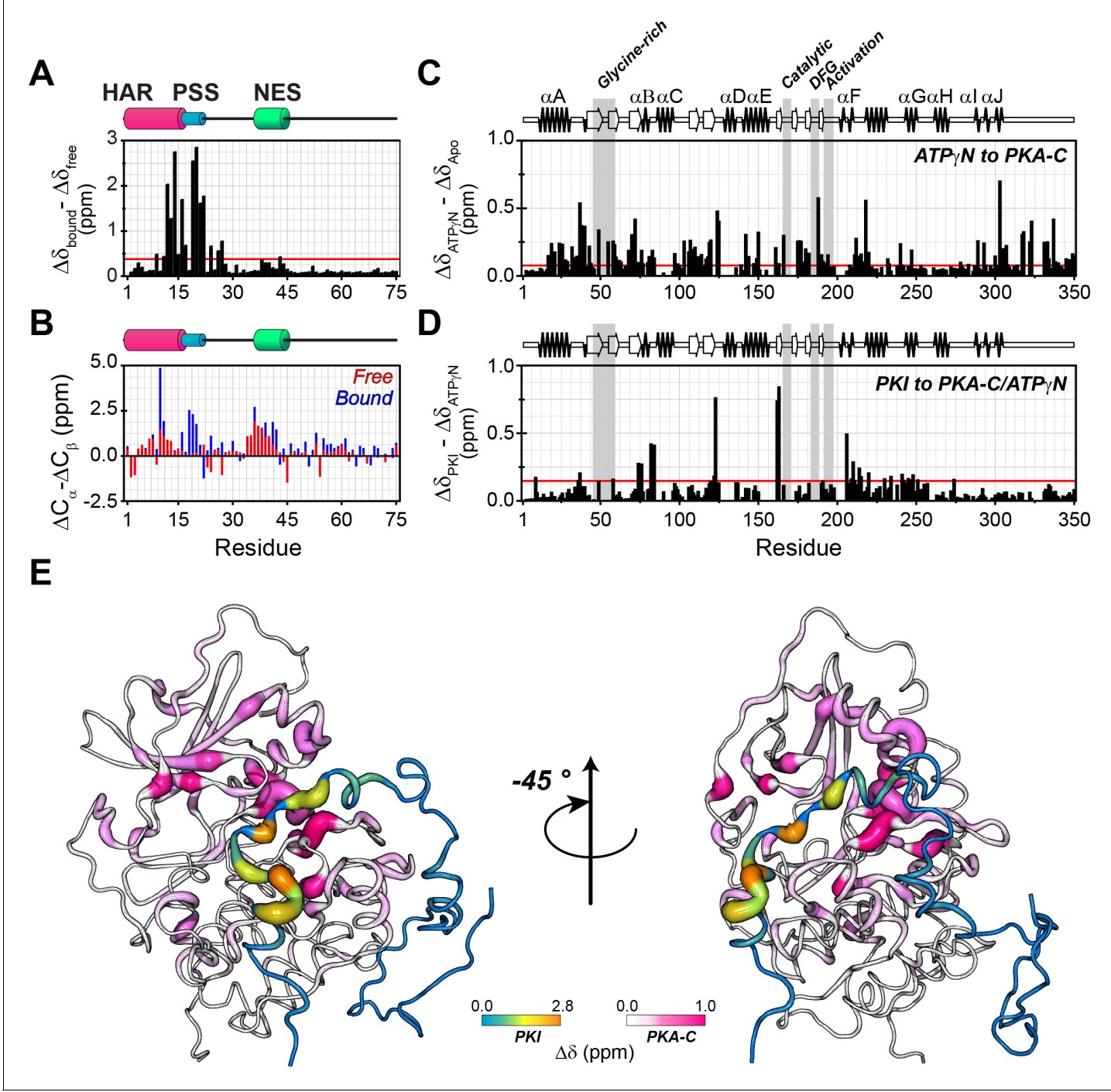

**Figure 3.** PKIα adopts partial secondary structure upon interaction with PKA-C. (**A**) Chemical shift perturbation (CSP) of the amide fingerprint of PKIα upon binding PKA-C/ATPγN binary complex calculated using *equation 1* (*Williamson, 2013*). The residues of PKIα that undergo significant CSPs are localized in the HAR and PSS, which are involved in strong electrostatic interaction within the binding site of PKA-C. (**B**) Chemical shift index (CSI) for Cα and Cβ of free (red) and bound (blue) PKIα. The PSS motif becomes more rigid upon interaction with PKA-C. The HAR and the NES adopt a transient α-helical conformation, which is enhanced upon binding the kinase. (**C**) CSP of PKA-C amide fingerprint upon binding ATPγN. (**D**) CSP of the PKA-C/ATPγN binary complex upon binding PKIα. The red lines in the histograms indicate one standard deviation from the average CSP: 0.38 ppm, 0.11 ppm, and 0.10 ppm for (**A**), (**C** and **D**), respectively. (**E**) CSPs for PKA-C and PKIα amide resonances mapped onto a selected conformer from the ensemble of the PKA-C/ATP/PKIα structures calculated from MD simulations (see Materials and methods).

The online version of this article includes the following figure supplement(s) for figure 3:

**Figure supplement 1.** Mapping PKA-C/PKIα interactions using [$^1$H,$^{15}$N] CCLS-HSQC experiments.

*Figure 3 continued on next page*

*Figure 3 continued*

**Figure supplement 2.** NMR dynamic parameters for free and bound PKIα.

binding PKIα, especially for αC, αF, αG helices as well as β6 loop. The map of the chemical shift changes on the PKA-C/ATPγN/PKIα complex for the two proteins is reported in *Figure 3E*. Interestingly, smaller changes are present for the NES region of PKIα, which are mirrored by higher values of heteronuclear nuclear Overhauser effects (NOE), indicating reduced motions of the backbone amides in the pico-to-nanosecond time scale (*Figure 4A*, *Figure 3—figure supplement 2B*; *Dyson and Wright, 2004*; *Bah et al., 2015*). The formation of the PKA-C/ATPγN/PKIα (ternary) complex is also reflected by the differences of longitudinal and transverse relaxation rates ($\Delta R_1$ and $\Delta R_2$) for both the HAR and PSS motifs upon forming the complex, which define a local rigidification of the PKIα backbone upon binding PKA-C, with the C-terminal portion (residues 47–75) remaining essentially unstructured (*Figure 4B,C*).

The $T_1/T_2$ ratios can be used as an estimate of the global correlation time for the PKA-C/PKI complex (*Kay et al., 1989*; *Cavanagh et al., 2007a*; *Cavanagh et al., 2007b*; *Gaspari and Perczel, 2010*). The PSS and HAR regions of PKIα, which show the most prominent chemical shift changes, adopt the overall correlation time of the kinase in agreement with previous fluorescence anisotropy measurements (*Figure 3—figure supplement 2C*; *Hauer et al., 1999b*). Interestingly, the $\Delta R_{ex}$ values obtained from the Carr-Purcell-Meiboom-Gill (CPMG) relaxation dispersion experiments show the presence of µs-ms motions for residues in the HAR and PSS motifs, which gradually decrease toward the NES motif, suggesting the presence of conformational interconversion of these regions upon binding (*Figure 4D*, *Figure 3—figure supplement 2D*). Next, we analyzed the large amplitude motions of the kinase-bound PKIα by conjugating MTSL to cysteine 59 in the C-terminal region (PKIα$^{S59C}$) and used the [$^1$H,$^{15}$N]-CCLS pulse sequence to simultaneously measure the PREs on both PKIα and PKA-C (*Olivieri et al., 2018*; *Figure 5A,B*). The most significant PREs for PKIα were detected for resonances of residues proximal to the MTSL label, with a gradual reduction moving away from residue 59. Interestingly, we detected some long-range effects for specific residues located in the HAR and PSS motifs and the intervening linker. This implies that the C-terminal

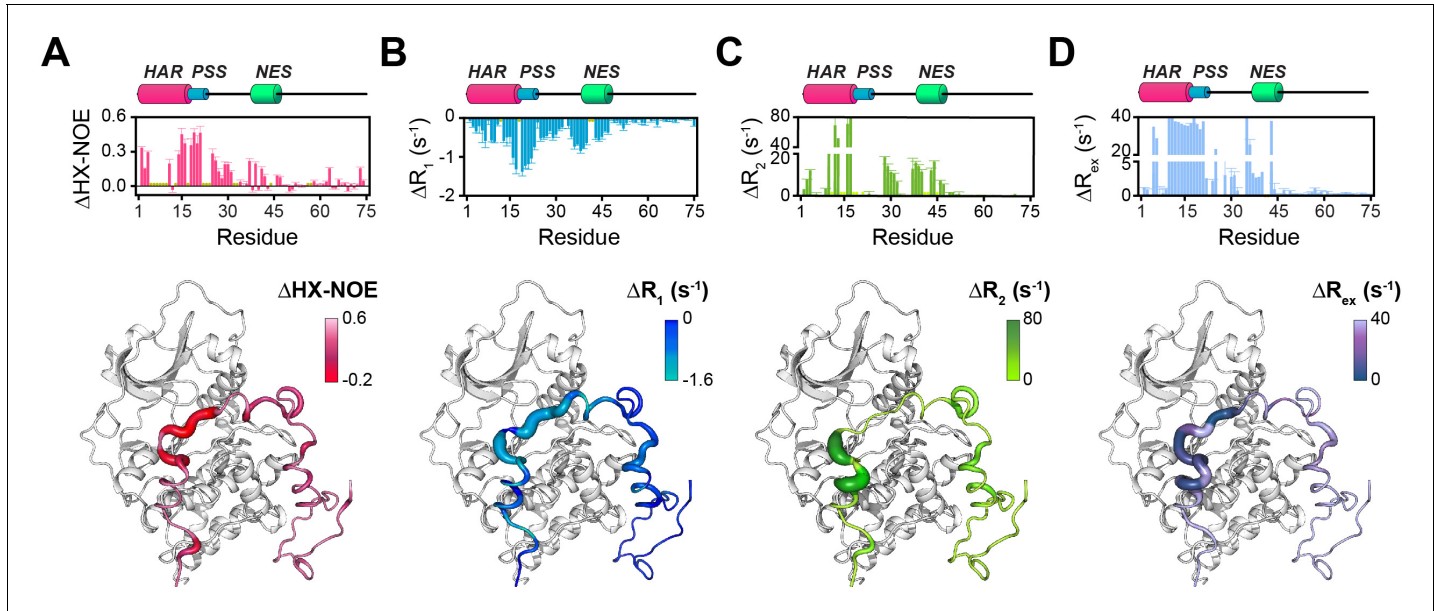

**Figure 4.** Changes in the NMR dynamic parameters of PKIα upon formation of the ternary complex with PKA-C and ATPγN. (**A**) Histogram of the changes in heteronuclear NOE (ΔHX-NOE) values of PKIα free and bound and map of the ΔHX-NOE values onto a selected conformer from the ensemble of the PKA-C/ATP/PKIα structures calculated from MD simulations (see Materials and methods). (**B-D**) Corresponding $\Delta R_1$, $\Delta R_2$, and $\Delta R_{ex}$ plots for PKIα free and bound forms. The sample concentration for free form of $^2$H/$^{15}$N PKIα was 0.2 mM; while for PKIα bound to PKA-C/ATPγN was 0.3 mM. All the experiments were recorded at 27˚C.

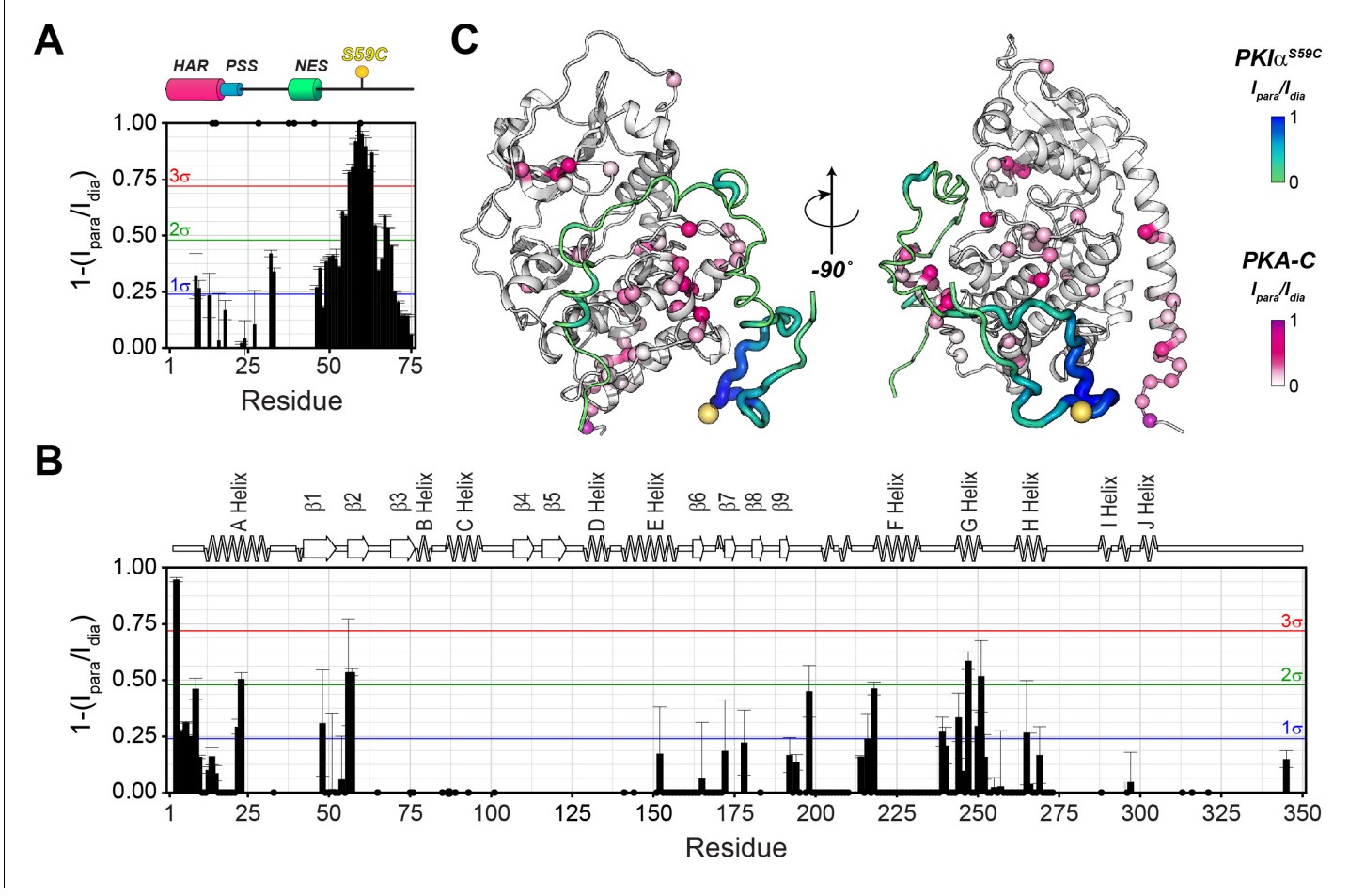

**Figure 5.** The C-terminal tail of PKIα interacts transiently with PKA-C. (**A**) Intermolecular $^1H_N$ PRE measurements between U-$^2$H/$^{13}$C/$^{15}$N PKIα$^{S59C}$ and U-$^2$H/$^{15}$N PKA-C/ATPγN. (**B**) Intermolecular PRE effects detected on PKA-C. (**C**) Intra- and inter-molecular PRE effects mapped onto the a selected conformer from the ensemble of the PKA-C/ATP/PKIα structures calculated from MD simulations. The yellow sphere illustrates where the spin label is attached. The black dots on both panel **A** and **B** indicate those resonances that broaden out beyond detection mainly in the sample with the spin-label active (paramagnetic sample). The horizontal lines indicate one (blue), two (green), and three (red) standard deviations from the mean (σ) of the observed PRE values. We refer as strong PRE effects values greater than 3σ, medium between 1 and 2σ, and weak less than 1σ.

domain of PKIα undergoes large amplitude motions and transiently interacts with the N-terminus, N-lobe (β₁-β₂ loop), C- lobe (G-, and H- helices) as well as the peptide-positioning loop of PKA-C. Taken together, the relaxation data show that PKA-C binding to PKIα causes differential effects on the various regions of the inhibitor. While *disorder-to-order* transitions occur for the HAR, PSS, and NES motifs, the C-terminal tail of the PKIα remains disordered. (*Figure 5C*).

## The PKA-C/PKIα complex relaxes through two structurally and kinetically distinct states to form the complex

To determine the mechanism of binding PKA-C and PKIα, we utilized stopped-flow rapid mixing fluorescence resonance energy transfer (FRET) and analyzed transient and total changes in donor fluorescence during the binding reaction. This procedure enables the interpretation of the kinetic mechanism and structural changes occurring during a binding reaction (*Muretta et al., 2015*; *Nesmelov et al., 2011*). To determine the kinetics of binding, we engineered a double mutant of PKA-C (PKA-C$^{C199A,S325C}$) and labeled the single reactive S325C with Alexa Fluor 488 (PKA-C$^{DONOR}$). We also expressed and purified three different mutants of PKIα (PKI$^{V3C}$, PKI$^{S28C}$, and PKI$^{S59C}$) and labeled each cysteine with tetramethylrhodamine-5-maleimide (PKI$^{ACCEPTOR-3}$, PKI$^{ACCEPTOR-28}$, and PKI$^{ACCEPTOR-59}$). For the first set of experiments, we mixed PKA-C$^{DONOR}$ with increasing concentrations of each acceptor (PKI$^{ACCEPTOR-3}$, PKI$^{ACCEPTOR-28}$, PKI$^{ACCEPTOR-59}$, (*Figure 6A–D*). As expected

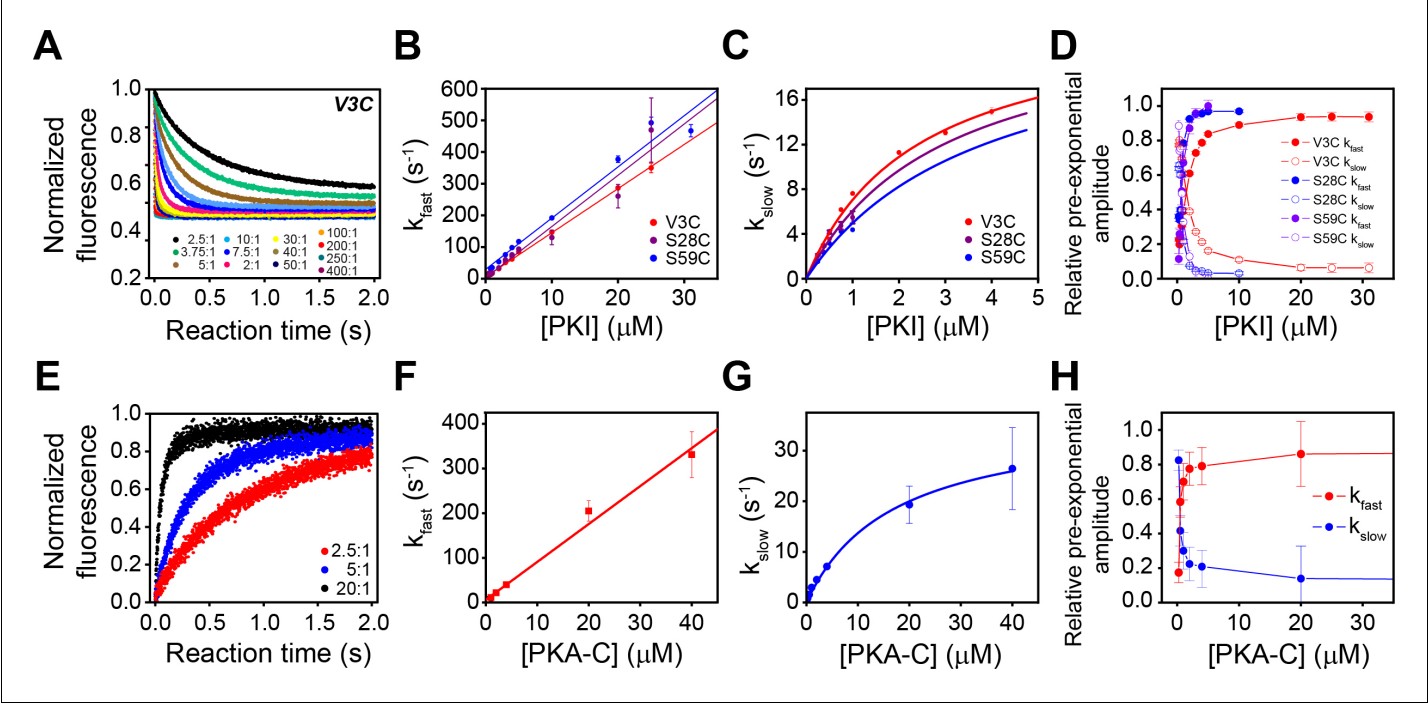

**Figure 6.** Multi-pathway mechanism of PKIα binding to PKA-C revealed by stopped-flow FRET experiments. (**A**) Total fluorescence of Alexa-488 labeled PKA-C$^{C199A,S325C}$ (100 nM) monitored at 520 ± 10 nm, mixed with varied concentrations of TMR labeled PKIα$^{V3C}$, ranging from 0 to 400 times the concentration of the enzyme. (**B-C**) Dependence of the fast and slow rate constants on the concentration of PKIα. (**D**) Relative pre-exponential amplitude obtained by fitting each transient to a bi-exponential function A1*Exp(-t*k$_{obs1}$) + A2*Exp(-t*k$_{obs2}$). (**E**) Total fluorescence of double-labeled PKIα$^{V3C, S59C}$ (Alexa-488, TMR) (100 nM) monitored at 520 ± 10 nm mixed at different concentrations of non-labeled PKA-C$^{C199A, S325C}$. (**F-G**) Dependence of the fast and slow rate constants on PKA-C concentration. (**H**) Relative pre-exponential amplitude obtained by fitting each transient to a bi-exponential function. Representative data from N = 3 experiments for each condition were repeated three times per experiment.

The online version of this article includes the following figure supplement(s) for figure 6:

**Figure supplement 1.** Time-resolved FRET data.

for the formation of the PKA-C/PKIα complex, we observed a gradual decrease in the total fluorescence of the donor probe. In a second set of experiments, we tested the time-dependence of long-range rearrangements of PKIα as suggested by PRE measurements (*Figure 6A*, *Figure 6—figure supplement 1A*). To this extent, we utilized the PKA-C$^{C199A,S325C}$ mutant without fluorescent label and engineered a double mutant of PKIα with two cysteines at residue 3 and 59 (PKI$^{V3C,S59C}$). the PKI$^{V3C,S59C}$ mutant was then labeled with both Alexa-488 (donor) and TMR (acceptor) at positions 3 and 59, respectively (PKI$^{DONOR-ACCEPTOR}$). When a fixed concentration of PKI$^{DONOR-ACCEPTOR}$ was titrated with increasing concentrations of unlabeled PKA-C$^{C199A,S325C}$, we detected an increase of the total fluorescence (*Figure 6E*, *Figure 6—figure supplement 1B*), indicating that the distance between the two probes of PKIα increases upon formation of the complex, that is the conformational ensemble of PKIα becomes on average more extended during the binding reaction in agreement with the PRE data. From an initial analysis of the binding curve, we found that the total fluorescence of the Alexa-488 donor emission from the PKA-C$^{DONOR}$/PKI$^{ACCEPTOR}$ changed bi-exponentially through the course of the binding reactions (*Figure 6A–D*). The bi-exponential behavior suggests the presence of an initial fast binding step followed by a slow structural rearrangement (*Table 1*; *Gianni et al., 2014*).

These two phases can be identified for all the PKI labeled species, irrespective of the fluorescence acceptor position. The rate constants for the fast phase increases linearly with PKIα concentration, with an average second-order rate constant of $1.63 \times 10^7$ M$^{-1}$ s$^{-1}$ and an extrapolated average dissociation rate constant of 7.0 s$^{-1}$. Comparatively, the observed rate constant for the slow phase increased hyperbolically with an average K$_{0.5}$ of 3.1 μM and a maximum value (k$_{slow}$) of 24 ± 1.8 s$^{-1}$.

**Table 1.** Rate constants obtained from the fluorescence decay of PKI$^{ACCEPTOR-3}$, PKI$^{ACCEPTOR-28}$, and PKI$^{ACCEPTOR-59}$ upon mixing with Alexa-labeled PKA-C$^{C199A,S325C}$ or the fluorescence buildup of PKI$^{DONOR-ACCEPTOR}$ upon mixing with unlabeled PKA-C$^{C199A,S325C}$ at 25˚C.

| | PKI$^{ACCEPTOR3}$ | PKI$^{ACCEPTOR28}$ | PKI$^{ACCEPTOR59}$ | PKI$^{DONOR-ACCEPTOR}$ |
|---|---|---|---|---|
| _Fast phase_ | | | | |
| $k_{on}$ (M$^{-1}$ s$^{-1}$) | $1.88 \pm 0.03 \times 10^7$ | $1.40 \pm 0.02 \times 10^7$ | $1.62 \pm 0.11 \times 10^7$ | $0.85 \pm 0.04 \times 10^7$ |
| $k_{off}$ (s$^{-1}$) | $13 \pm 3$ | $5 \pm 1$ | $2 \pm 10$ | $6 \pm 7$ |
| $K_d^{app}$ (nM) | $700 \pm 170$ | $390 \pm 110$ | $150 \pm 620$ | $700 \pm 800$ |
| _Slow phase_ | | | | |
| $k_{slow}$ (s$^{-1}$) | $24 \pm 2$ | ND | ND | $37 \pm 2$ |
| $K_{0.5}$ (µM) | $2.4 \pm 0.4$ | $3.0 \pm 0.4$ | $3.8 \pm 0.5$ | $17 \pm 3$ |

ND: Non determined.

We also observed two distinct binding phases for the second set of experiments with unlabeled PKA-C$^{C199A,S325C}$ and PKI$^{DONOR-ACCEPTOR}$. Indeed, the kinetic constants obtained for these experiments are very similar to those found in the first set of the experiments, underscoring the same binding mechanisms. The pre-exponential amplitudes for the fast phases increased hyperbolically, while the slow phase decreased hyperbolically (*Figure 6D–H*). Taken together, the behavior of the rate constants and the pre-exponential amplitudes for the fast and slow phases are consistent with a multi-pathway mechanism in which there is a fast binding phase of PKIα ensembles competent for binding and a subsequent structural rearrangement of these conformers upon binding (*Dogan et al., 2014*; *Gianni et al., 2014*).

## Conformational landscape and kinetics of conformational transitions of PKIα

Using chemical shift (CS) and residual dipolar couplings (RDC) (*Figure 2D*) as structural restraints, we performed replica-averaged metadynamics (RAM) to define the energy landscapes of free and kinase-bound PKIα. During the simulations, the HAR and NES motifs of free PKIα undergo multiple coil-to-helix transitions (*Figure 8*). We identified two major basins, featuring a compact ensemble ($R_g$ <1.5 nm), with three relative minima with helical structures for the HAR and NES motifs; and a more extended ensemble ($R_g$ >3.0 nm), where the protein is essentially disordered. The HAR motif populates mostly a random coil conformation, with a sparsely populated helical conformation, whereas the NES motif populates both helical and coil conformations (*Figure 7A,B*). To validate the RAM-generated ensembles, we carried out small angle X-ray scattering (SAXS) experiments and back-calculated the SAXS profiles from the conformers sampled in the RAM trajectories. *Figure 7C* shows the scattering intensity plots (log *I(q)* vs *q*) and the corresponding Kratky plot for free PKIα. These profiles show the typical signatures of intrinsically disordered proteins, with the Kratky plot featuring a plateau at high *q* values as well as the absence of a bell-shaped curve (*Figure 7C*, bottom panel). In addition, the *P(r)* curve shows that PKIα adopts a highly extended conformational ensemble, with an abnormally large D$_{max}$ (~110 Å) and R$_g$ (~30 Å) values for a protein of its size, both parameters are indicative of an extended, intrinsically disordered protein (*Kikhney and Svergun, 2015*; *Figure 8D*). In fact, the back-calculated SAXS profiles from the RAM-generated ensembles are in excellent agreement with the experimental SAXS scattering profiles ($\chi$ = 0.86 for q < 0.2) (*Figure 7C*). In the bound state, the HAR motif of PKIα is more ordered and adopts a more stable helical conformation; whereas the NES motif is considerably more dynamic and adopts a transient helical conformation (*Figure 7E*).

In this case, the SAXS data are less informative as the Kratky plot for the PKA-C/ATPγN/PKIα complex is dominated by the bell-shaped curve of the globular enzyme, with a well-defined maximum in the low *q* region (~0.08 Å$^{-1}$) that decays to near zero at high *q* values. To determine the rates of conformational exchange and the populations of PKIα free and bound, we performed unbiased molecular dynamics (MD) simulations and analyzed the conformational transitions building a Markov model. To simplify and visualize the conformational space of PKIα, we reduced the high-dimensionality space of the residue-residue contacts into a 6-dimensional space using time-lagged

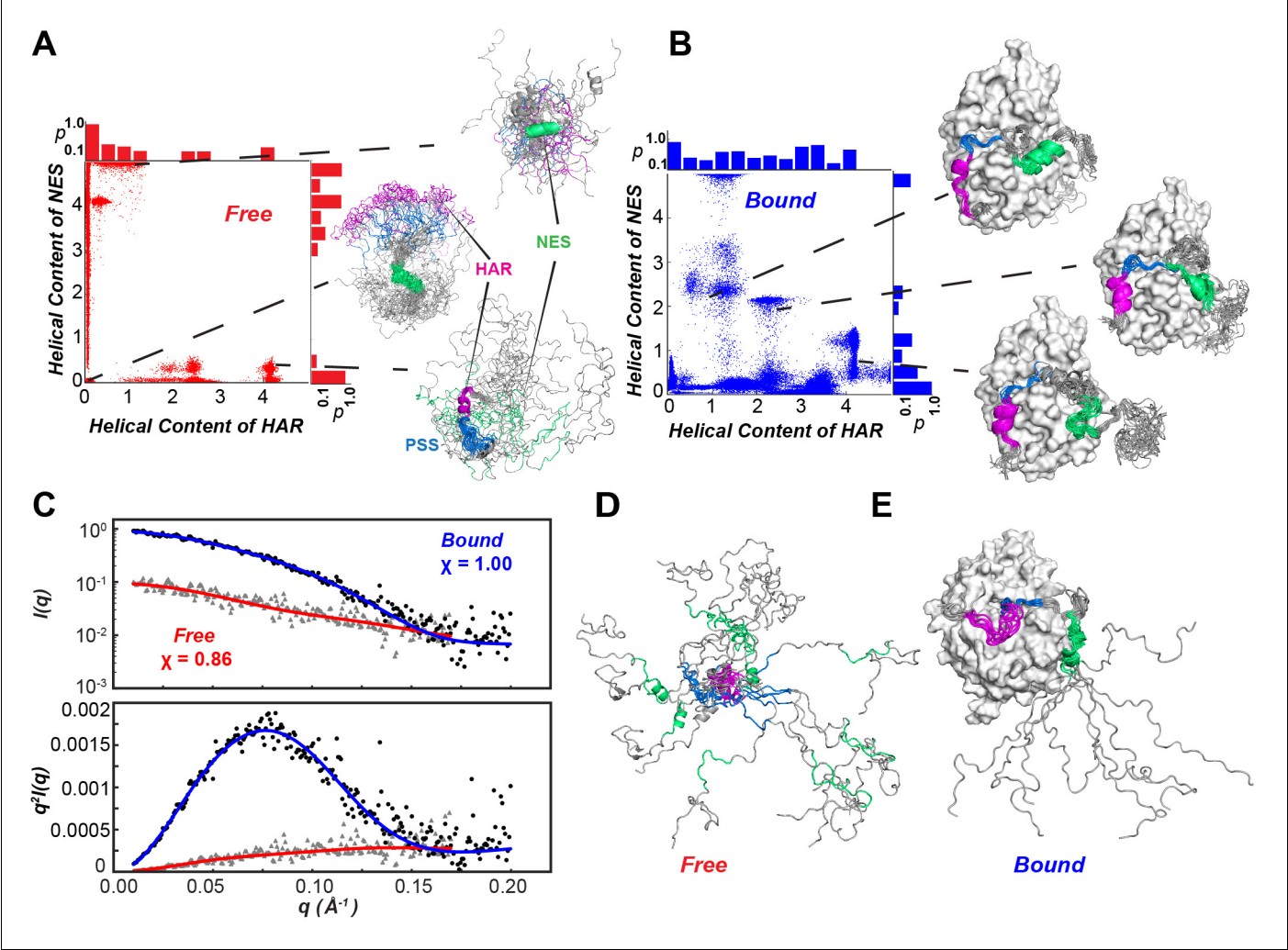

**Figure 7.** Conformational ensembles free and bound of PKIα by SAXS and metadynamics. (A-B) Distribution of helical populations for the HAR and NES regions in the free (A) and bound (B) PKIα ensembles. (C) SAXS data obtained with PKIα free and the PKA-C/PKIα complex in the presence of ATPγN. The top panel shows the scattering intensity, *I(q)*, as a function of q and the bottom panel the Kratky plot. The blue (complex) and red (free PKIα) traces are the best-fit curves obtained by fitting predicted scattering data from the conformation ensembles from RAM simulations. (D) Conformational ensemble of free PKIα that best fits the SAXS profile. E. PKIα ensemble bound to PKA-C that best fits the SAXS profile.

independent component analysis (tICA). We then projected the free energy landscape along the main components, tIC1 and tIC2. In this reference frame, free PKIα occupies six local minima (ΔΔG < 1 kcal/mol) and a higher energy state (ΔΔG ~ 2.5 kcal/mol) (*Figure 8*). We then clustered the conformers into four major ensembles based on the secondary structure content of the HAR, PSS, and NES motifs: HLH (helix-loop-helix), CLC (coil-loop-coil), HLC (helix-loop-coil), and CLH (coil-loop-helix) (*Figure 8—figure supplement 1*). For free PKIα, the CLH ensemble is dominant, with a mean population of 51%, followed by CLC (32%), HLC (15%), and HLH (~2%) (*Figure 8A*). The kinetics of the conformational transitions were calculated using the mean-first-passage-time (MFPT) among these states. We found that all the conformational transitions take place on a μs time scale, with the slowest transitions from the HLC, CLH, and CLC ensembles to the HLH ensemble; while the transitions from CLC, HLH, and HLC to the CLH ensemble are more rapid (*Figure 8*, *Figure 8—figure supplement 1*). For PKIα bound to PKA-C, the conformational space spanned becomes more restricted (*Figure 8D*). Specifically, the HLC, CLC, and HLH minima become more populated, with 72, 22, and 6% (*Figure 8B*). In addition, the transition kinetics from CLC and HLC to the HLH ensemble is estimated to occur approximately 10 times faster (*Figure 8*, *Figure 8—figure supplement 1*).

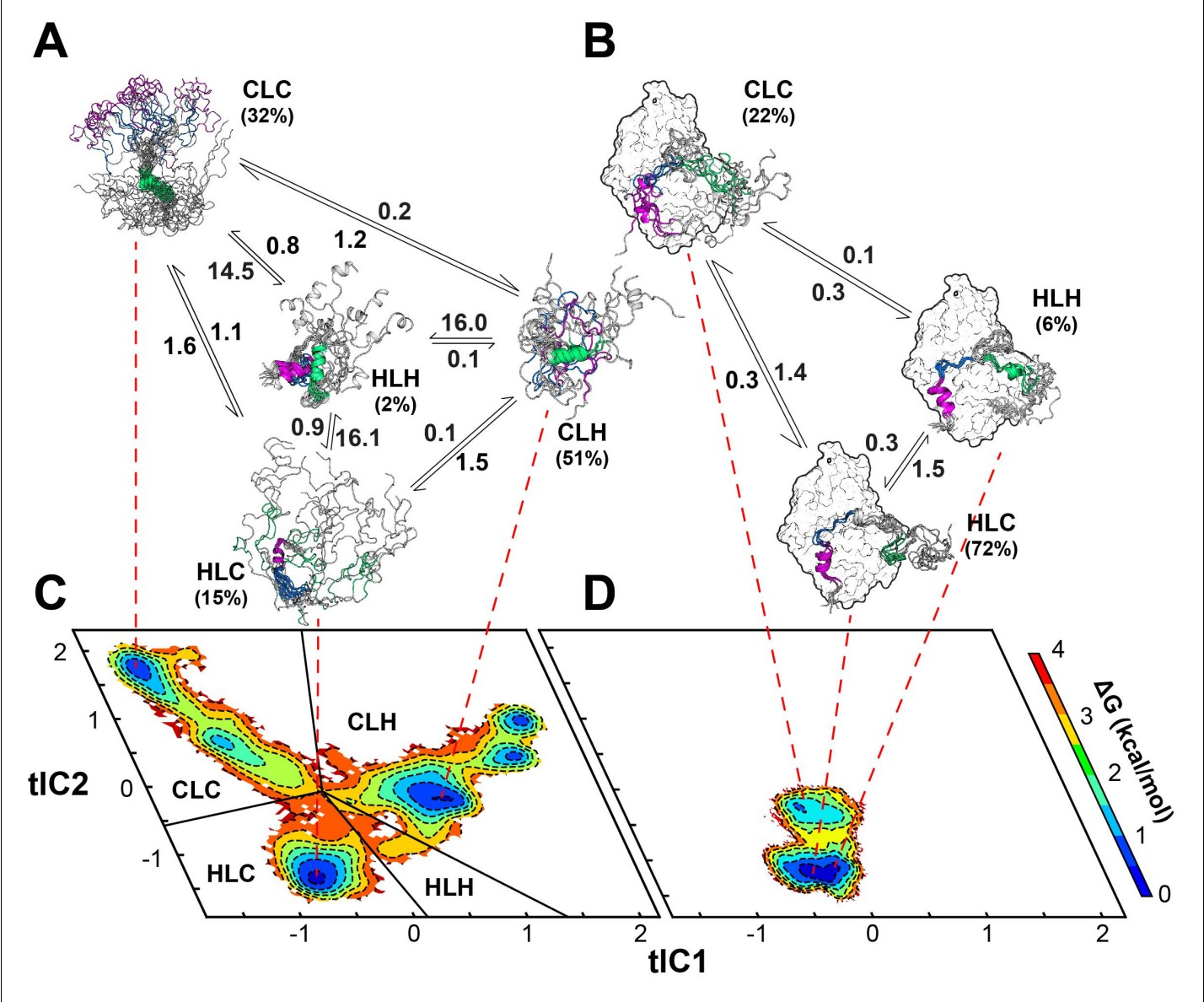

**Figure 8.** Multi-state binding kinetics of PKI α to PKA-C from Markov model. (**A**) Schematic illustration of the transition kinetics between different conformational ensembles of PKIα in the free form. (**B**) Corresponding scheme for the bound ensembles. The population of each state is indicated as percentage. The kinetic constants expressed in microseconds are indicated on the arrows. (**C**) 2D free energy landscape for PKIα in the free form projected along the first two tICs (**D**) Corresponding free energy landscape for PKIα in the bound forms. The major conformational ensembles of free PKIα in the free form can be categorized into four states, featuring different combination of helix/coil/loop for the HAR, PSS and NES motifs. The online version of this article includes the following figure supplement(s) for figure 8:

**Figure supplement 1.** Scatter plot from MD simulations showing the conformational ensembles of PKIα in the free and bound forms projected along the first two tICs.

## Proposed mechanistic model

Based on NMR, SAXS, fluorescence, RAM simulations as well as Markov model analysis, we propose that PKIα binding to PKA-C occurs via a multistate recognition pathway. We envision that free PKIα spans a broad conformational space with incipient helical elements for the HAR and NES motifs. These conformations are in fast exchange in the NMR time scale. We propose that the initial step (fast) of recognition by PKA-C is driven by electrostatic interactions between the conformations of PKIα featuring an exposed PSS motif (CLC and HLC) and PKA-C (*Tsigelny et al., 1996*). These

encounter complexes undergo slower conformational rearrangements in which a) the HAR motif folds upon binding in a helical conformation, b) the PSS motif binds within the active cleft in an extended conformation, and c) the transient helix of the NES motif adopts a more defined conformation. To validate our hypothesized mechanism, we fitted the fluorescence curves using a set of equations detailed in the supporting information (*Scheme 1* and *Scheme 2*). Under these constraints, we obtained a kinetic constant of $1.5 \times 10^7$ M$^{-1}$s$^{-1}$ for the formation of the PKA-C*/PKI* encounter complex and a slow relaxation of the complex with a constant of 2400 s$^{-1}$. These kinetics parameters are in the same order of magnitude of the association rates measured for binding of other intrinsically disordered proteins (*Mollica et al., 2016*).

## Discussion

When unleashed from the R-subunit, the C-subunit of PKA is in a constitutively active state, where binding of nucleotide engages the N- and C-lobes priming the kinase for substrate recognition and catalysis. R-subunits and PKI share small linear motifs (SLiMs) that bind the substrate binding site with high affinity and trap PKA-C in an inactive state. SLiMs are intrinsically disordered regions that are part of the allosteric machinery of kinases and control the cross-talk between different kinase signaling pathways (*Akimoto et al., 2014*; *Akimoto et al., 2013*; *Delaforge et al., 2018*; *Gógl et al., 2019*; *Kragelj et al., 2015*; *Wang et al., 2011*; *Wright and Dyson, 2015*; *Guiley et al., 2019*). PKI can be portrayed as a prototypical SLiM that is able to bind and inhibit the kinase and recruit the CRM1/RanGTP complex to export the enzyme out of the nucleus, thereby regulating cell proliferation. Previous structural studies focused on the 5–24 region of PKIα that encompasses both the HAR and PSS motifs, which has been instrumental in trapping the ternary complex of the kinase and obtain the first crystal structure of a kinase (*Knighton et al., 1991a*). Subsequent spectroscopic studies provided evidence for the dynamic nature of the PKIα peptide, suggesting the presence of minimal secondary structural elements such as the HAR and NES regions, with a C-terminal tail completely disordered (*Padilla et al., 1997*; *Hauer et al., 1999a*). The HAR is extremely important for binding/anchoring to the catalytic subunit as well as positioning the PSS within the catalytic cleft of the enzyme (*Walsh et al., 1971*; *Cheng et al., 1986*). However, most PKA target substrates do not possess this motif and the only substrate recognition requirement is the highly positively charged consensus sequence in the linear PSS motif. In fact, in most cases this recognition motif is located in highly dynamic or intrinsically disordered regions as it molds into the binding groove of the kinase situated at the interface between the small and large lobes. Our studies bring new insights into the dynamic nature of the PKIα recognition mechanism. After the initial binding of ATP in the pocket between the two lobes that primes the substrate binding site (*Chu et al., 2017*; *Hyeon et al., 2009*; *Masterson et al., 2010*), this highly disordered polypeptide, with incipient helices at both the HAR and NES motifs, is driven toward the kinase via positively charged Arg residues in the PSS sequence. Deletion of one of the Arg residues in the PSS provokes a dramatic reduction of the phosphorylation kinetics and has been linked to the progression of dilated cardiomyopathy (*Kim et al., 2015*; *Haghighi et al., 2006*). On the other hand, mutations in the active site of the kinase that change electrostatic interactions such as L205R dramatically influence the

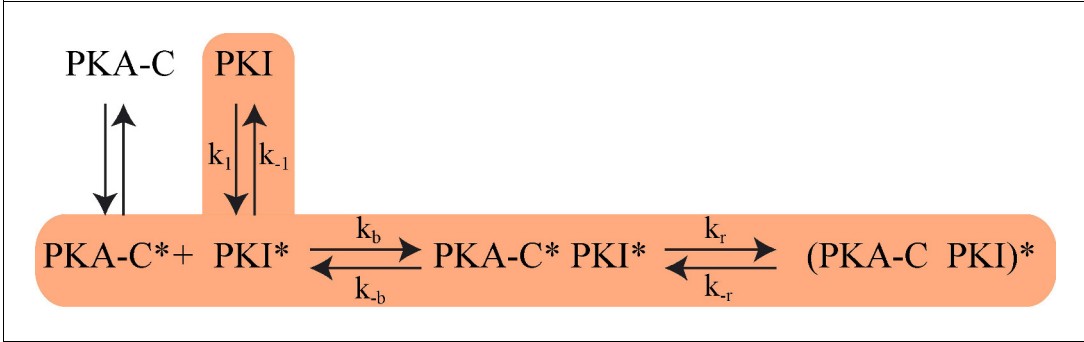

**Scheme 1.** The proposed Kinetic Model of PKA-C/PKIα binding. Highlighted pathway is used in the numerical fitting of Stopped Flow rapid mixing FRET data using MATLAB.

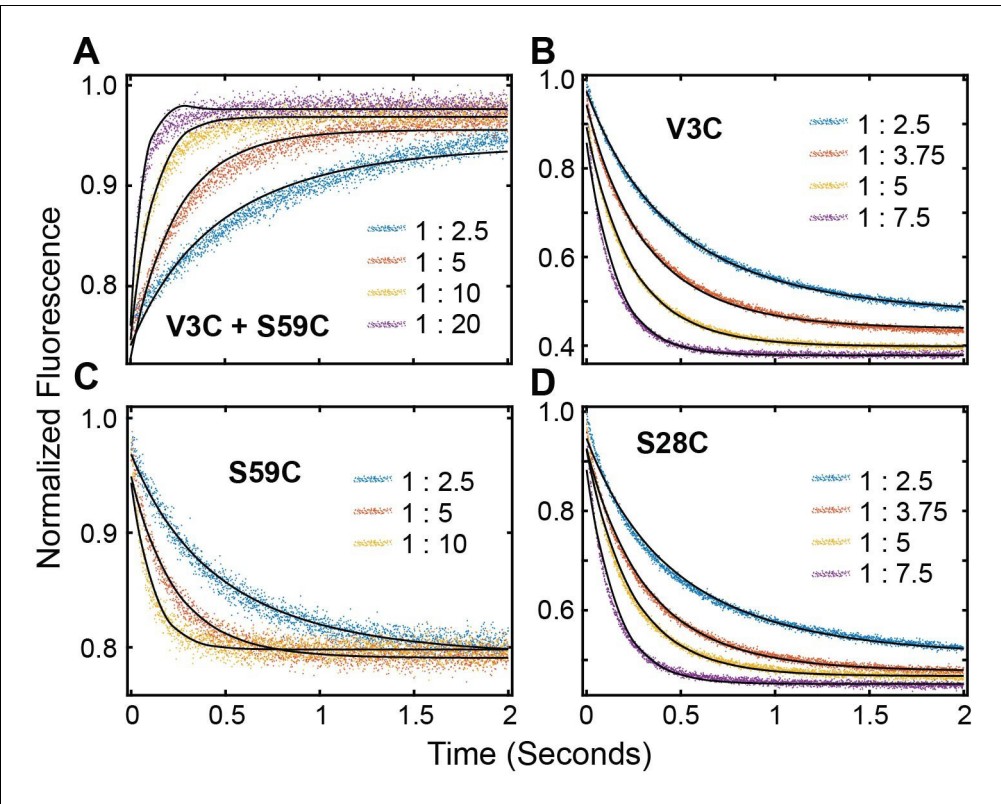

**Scheme 2.** FRET Profiles of (A) double labeled PKI, (B) PKI[ACCEPTOR-3], (C) PKI[ACCEPTOR-59] and (D) PKI[ACCEPTOR-28] fitted with kinetic model in *Scheme 1*.

thermodynamics and kinetics of substrate binding, leading to the development of adrenocortical adenomas and the resultant Cushing's syndrome (*Calebiro et al., 2014*; *Cheung et al., 2015*; *Walker et al., 2019*). More importantly, our studies reveal a significant ordering of the PKIα structure upon interaction with the kinase, particularly for the HAR, PSS, and NES motifs. This ordering occurs in the second, slower binding step as revealed by FRET measurements, while the C-terminal tail of PKIα remains essentially disordered, undergoing large amplitude motions with transient interactions with the C-lobe of the kinase. Interestingly, when PKIα is bound to PKA-C, the NES region undergoes a structural rearrangement, showing a higher degree of helicity as supported by NMR and MD simulations. The NES motif comprises Leu 37, Leu 41, and Leu 44 that are essential for high binding affinity to the nuclear export machinery CRM1/RanGTP (*Güttler et al., 2010*; *Fu et al., 2018*). The CRM1 binding pocket is a relatively rigid cavity and NES sequences must possess significant plasticity to mold into the binding pocket (*Güttler et al., 2010*; *Fu et al., 2018*). This observation could explain why the NES assumes a defined conformation only upon interaction with CRM1. Based on our data, it is possible to speculate that the interactions of PKIα with PKA-C preambles the formation of the complex and that the helix-coil equilibrium of the NES domain is critical for recruiting the CRM1/RanGTP nuclear export complex (*Fu et al., 2018*; *Fung et al., 2017*; *Güttler et al., 2010*). This complex binding mechanism shows how SLiMs can orchestrate multiple functions and regulate kinases through disordered ancillary proteins or protein domains.

# Materials and methods

**Key resources table**

| Reagent type (species) or resource | Designation | Source or reference | Identifiers | Additional information |
|---|---|---|---|---|
| Gene (*Oryctolagus cuniculus*) | PKIA or PKIα | | | Uniprot ID P61926 |
| Gene (*Mus musculus*) | PKA-CA or PKA-C | | | Uniprot ID: Q9DBC7 |
| Strain, strain background (*Escherichia coli*) | BL21(DE3) | New England Biotech (NEB) | C2527I | Chemically competent cells |
| Recombinant DNA reagent | pT7-7 PKIα | Dr. Herberg (Universität Kassel, Germany) | | DOI: 10.1042/BJ20071665 |
| Recombinant DNA reagent | PKIα$^{V3C}$ | This study | | Single Cys mutant of PKIα |
| Recombinant DNA reagent | PKIα$^{S28C}$ | This study | | Single Cys mutant of PKIα |
| Recombinant DNA reagent | PKIα$^{S59C}$ | This study | | Single Cys mutant of PKIα |
| Recombinant DNA reagent | PKIα$^{V3C, S59C}$ | This study | | Double Cys mutant of PKIα |
| Recombinant DNA reagent | PKA-Cα | Prof. Taylor S.S. (USCD, CA, USA) | | (*Hemmer et al., 1997*) DOI: 10.1006/abio.1996.9952 |
| Recombinant DNA reagent | (His$_{6X}$)-PKA-RIIα$^{R213K}$ | Prof. Taylor S.S. (USCD, CA, USA) | | (*Hemmer et al., 1997*) DOI: 10.1006/abio.1996.9952 |
| Sequence-based reagent | PKIα$^{V3C}$ | This study | | PCR primer (Forward): aaggagatatacat atgggaactgattgcg aaactacttatgccgatttta |
| Sequence-based reagent | PKIα$^{S28C}$ | This study | | PCR primer (Forward): ccatccacgatatc ctggtctgcagtgcttccgg |
| Sequence-based reagent | PKIα$^{S59C}$ | This study | | PCR primer (Forward): aggaagatgctcaaagatctt gcactgaacaatccggagaag |
| Sequence-based reagent | PKIα$^{V3C, S59C}$ | This study | | PCR primer (Forward): 1)aaggagatatacatatggg aactgattgcgaaa ctacttatgccgatttta 2)aggaagatgctcaaagatct tgcactgaacaatccggagaag |
| Chemical compound, drug | MTSL | Toronto Research Chemical | O875000 | Spin label |
| Chemical compound, drug | dMTSL | Toronto Research Chemical | A188600 | Spin label |
| Chemical compound, drug | Alexa Fluor 488 C5 Meleimide | Thermo Fisher Scientific | A10254 | FRET acceptor |
| Chemical compound, drug | Tetramethylrhodamine-5-maleimide (TMR) | Life Technologies | T6027 | FRET donor |
| Chemical compound, drug | AMP-PNP or ATPγN | Roche Applied Science | 10102547001 | ATP analogous |
| Commercial assay or kit | QuikChange Lightning Multi Mutagenesis Kit | Agilent genomics | 210519 | Commercial mutagenesis kit |

*Continued on next page*

*Continued*

| Reagent type (species) or resource | Designation | Source or reference | Identifiers | Additional information |
|---|---|---|---|---|
| Commercial assay or kit | PepTag Assay, Non-Radioactive Detection of PKA | Promega | V5340 | Commercial assay kit |
| Software, algorithm | TopSpin 3.0 | Bruker Inc | | https://www.bruker.com/ |
| Software, algorithm | NMRFAM-Sparky | NMRFam | | https://nmrfam.wisc.edu/nmrfam-sparky-distribution/ |
| Software, algorithm | NMRPipe | Delaglio F, NIH (*Delaglio et al., 1995*) | | https://www.ibbr.umd.edu/nmrpipe/install.html |
| Software, algorithm | PyMol | Schrödinger, LLC | | https://pymol.org |
| Software, algorithm | MatLab2019b | MathWorks | | https://www.mathworks.com/products/matlab.html |
| Software, algorithm | GraphPad Prism 8 | GraphPad Software Inc | | https://www.graphpad.com/ |
| Software, algorithm | Origin 8 | OriginLab | | https://www.originlab.com/ |
| Software, algorithm | SAXSQuant software suit | Anton Paar | | N/A |
| Software, algorithm | Primus | ATSAS 2.8.3 software | | https://www.embl-hamburg.de/biosaxs/download.html |
| Software, algorithm | MultiFoXS server | (*Schneidman-Duhovny et al., 2013*). | | https://modbase.compbio.ucsf.edu/multifoxs/ DOI: 10.1016/j.bpj.2013.07.020 |
| Software, algorithm | GROMACS 4.6 | (*Hess et al., 2008*) | | http://www.gromacs.org/ |
| Software, algorithm | PLUMED 2.1 | (*Bonomi et al., 2009*) | | https://www.plumed.org/doc-v2.5/user-doc/html/_c_h_a_n_g_e_s-2-1.html |
| Software, algorithm | ALMOST 2.1 | (*Fu et al., 2014*) | | https://sourceforge.net/projects/almost/ |
| Software, algorithm | MSMBuilder 3.5 | (*Harrigan et al., 2017*) | | http://msmbuilder.org/3.5.0/index.html |
| Software, algorithm | δ2D | (*Camilloni et al., 2012*) | | http://www-mvsoftware.ch.cam.ac.uk/index.php/d2D |

## Expression and purification of Full-length PKIα

Gene coding for full-length PKIα (*Oryctolagus cuniculus, PKIA PRKACN1*) subcloned into pT7-7 vector was a gift to us by Dr. Friedrich Herberg (Universität Kassel, Germany). For the fluorescence and PRE-experiments, three different single-site (PKIα$^{V3C}$, PKIα$^{S28C}$ and PKIα$^{S59C}$) and a double-site (PKIα$^{V3C, S59C}$) mutants were generated by QuikChange Lightning mutagenesis kit (Agilent genomics) using PKIα as template. The choice of the sites of mutation was based on previous work (*Hauer et al., 1999b*). Recombinant full-length PKIα was expressed in *E. coli* BL21 (DE3) cells at 30°C and the purification was performed as previously reported (*Hauer et al., 1999a*). Briefly, transformed cells were grown in Luria-Bertani (LB) medium for fluorescence while uniformly labeled $^{15}$N, $^{13}$C/$^{15}$N, and $^{2}$H/$^{13}$C/$^{15}$N PKIα was expressed in minimal medium (M9) for NMR experiments. Protein expression was induced by the addition of 0.4 mM of IPTG to a culture with optical density of ~1.3–1.4 at 30°C and was performed for 5 hr. The cell pellet was resuspended in 20 mM MOPS (pH 7.5) and lysed through sonication. The lysate was then centrifuged at 20,000 rpm for 30 min. The supernatant was then heated in a boiling water bath at 95°C for 5 min to precipitate out most of the undesired proteins. The protein suspension was centrifuged a second time at 20,000 rpm for 30 min. The resulting supernatant was dialyzed against 20 mM TRIS-HCl (pH 7.0) buffer overnight. A second purification step with anion exchange chromatography was carried out using HiTrap Q HP column (GE Healthcare Biosciences Corp) in 20 mM TRIS at pH 7.0 where a NaCl gradient was used to elute the

protein. A reversed phase HPLC purification step was also performed to obtain a higher grade of purity using C18 column (HiCHROM, UK) using water as buffer A and isopropanol as buffer B with 0.1% trifluoroacetic acid in both buffers. The purified peptide was concentrated, lyophilized, and stored at −20°C. The final product was accessed using SDS-PAGE with a final purity of >97%. The molecular weight and the quantity of the peptide were verified by amino acid analysis (Protein chemistry laboratory at Texas A and M University, TX, USA). All the cysteine mutants of PKIα were purified following the same protocol used for the wild-type protein with additional supplement of reducing agent (5 mM β-mercaptoethanol) in all the dialysis buffers.

## Expression and purification of PKA-C

Recombinant catalytic subunit of PKA-C (*Mus musculus* gene) was expressed in *E. coli* BL21 (DE3) cells at 24°C in M9 minimal medium. The purification was performed using the His6-RIIa(R213K) subunit as described previously (*Hemmer et al., 1997*). A subsequent second purification step was performed using a HiTrap SP HP cation exchange column (GE Healthcare Biosciences Corp) to separate out the three isoforms of PKA-C that differ in their phosphorylation profiles (*Yonemoto et al., 1993*). The purified protein was then stored in phosphate buffer supplied of 10 mM DTT, 10 mM MgCl$_2$, and 1.0 mM NaN$_3$, at 4°C. For isotopically $^2$H-labeled protein, the cells were growth 80% D$_2$O M9 minimal medium in a bench fermenter (2.0 L). The most abundant isoform of PKA-C, corresponding to phosphorylation at S338, T197, and S10 residues (isoform II) (*Walsh and Ashby, 1973*), was used for all experiments. The purity was assessed using SDS-PAGE electrophoresis and the final purity was >97%. The kinase activity was tested with a gel-shift assay from Promega (Fitchburg, Wisconsin) and quantified using A$_{280}$ = 52,060 M$^{-1}$ cm$^{-1}$.

The C199A S325C mutant of the catalytic subunit of PKA (PKA-C$^{C199A, S325C}$) was recombinantly expressed by growing transformed *E. coli* BL21(DE3) cells in LB medium and induced by addition of IPTG at 24°C as described previously. The mutant PKA-C was purified using the previously described protocol (*Masterson et al., 2009*). Briefly, cells were resuspended in lysis buffer (30 mM MES, 1 mM EDTA, 50 mM KCl, 5 mM β-mercaptoethanol, 0.15 mg/mL lysozyme, 1 mM PMSF, pH 6.5), and lysed using French press at 1000 psi. The supernatant after pelleting the cell debris was batch-bound with P11 phosphocellulose resin in lysis buffer at pH 6.5. PKA-C was purified by eluting over a gradient of 0–500 mM KH$_2$PO$_4$. The different isoforms of PKA-C were further separated by cation exchange chromatography with a HiTrap SP column using a KCl gradient in 20 mM KH$_2$PO$_4$ at pH 6.5. Upon purification, wild-type or mutant proteins were supplied with 10 mM MgCl$_2$, 10 mM DTT and 1 mM NaN$_3$ and stored at 4°C.

## Spin labeling of proteins

Lyophilized single site $^{15}$N PKIα mutants were first dissolved in 1 mL of buffer containing 20 mM KH$_2$PO$_4$, pH 7.0, and incubated for about an hour in the presence of 1 mM TCEP. The sample was concentrated to 0.6 mM. The protein solution was then buffer exchanged into 20 mM KH$_2$PO$_4$, 90 mM KCl, 10 mM MgCl$_2$, 1 mM NaN$_3$ pH 7.0. The sample was concentrated and divided into two equal aliquots to perform two separate spin labeling reactions in parallel: one using the paramagnetic spin label MTSL (1-oxyl-2,2,5,5-tetramethyl- δ−3-pyrroline-3-methyl)methanethiosulfonate, Toronto Research Chemicals, Inc), and the second using the diamagnetic form, dMTSL (1-acetyl-2,2,5,5-tetramethyl-δ−3-pyrroline-3-methyl)methanethiosulfonate, Toronto Research Chemicals, Inc). The reactions were carried out at 4°C overnight with ten-fold excess of labeling compounds that was then removed by passing the reaction solution through a 5 mL HiTrap Desalting column (GE Healthcare Biosciences Corp.) equilibrated with 20 mM KH$_2$PO$_4$, 90 mM KCl, 10 mM MgCl$_2$, 1 mM NaN$_3$, pH 6.5. The extent of labeling was assessed by ESI TOF mass spectrometry (Mass Spectrometry Laboratory, Department of Chemistry, University of Minnesota), and was found to be >99%.

## NMR sample preparation

The NMR samples used to study the free form of PKIα were composed primarily by $^{15}$N and $^{13}$C/$^{15}$N labeled protein expressed in M9 media. The final concentration of samples ranged from 0.5 mM to 0.8 mM in 20 mM KH$_2$PO$_4$, 90 mM KCl, 10 mM DTT, 10 mM MgCl$_2$, 1 mM NaN$_3$ at pH 6.5. Samples for the assignment of the PKIα in complex with PKA-C were performed using uniformly $^2$H/$^{13}$C/$^{15}$N and $^2$H/$^{15}$N labeled PKIα and uniformly $^2$H/$^{15}$N PKA-C. The ternary complex between PKIα and

PKA-C and ATP-analogues (AMP-PNP) were reconstituted in 20 mM $KH_2PO_4$, 90 mM KCl, 12 mM of ATPγN, 10 mM DTT, 10 mM $MgCl_2$, 1 mM $NaN_3$ at pH 6.5. A molar ratio of 1:1.2 (PKIα:PKA-C) to saturate the complex with a concentration of 0.150 mM of PKIα. All experiments were performed at 27°C. The denaturation experiments were carried out on 0.3 mM sample of uniformly $^{15}$N PKIα resuspended in 20 mM $KH_2PO_4$, 90 mM KCl, 10 mM DTT, 10 mM $MgCl_2$, 1 mM $NaN_3$, 8 M Urea at pH 6.5. The $^{1}H_N$-Γ2 relaxation experiments were performed on 0.25 mM $^{15}$N labeled single mutants of PKIα for the free form and uniformly $^{2}$H/$^{13}$C/$^{15}$N protein for the bound form. In both preparations the buffer used for the analysis was 20 mM $KH_2PO_4$, 90 mM KCl, 10 mM $MgCl_2$, 1 mM $NaN_3$ at pH 6.5. For all the NMR experiment carried out on the PKA-C/ATPγ/PKIα complex, a low protein concentration sample was prepared to due to the high tendency to aggregate.

## Preparation of fluorescent labeled proteins

Lyophilized single-site PKIα mutants were first dissolved in 1 mL of buffer containing 20 mM $KH_2PO_4$, pH 7.0, and incubated for about an hour in the presence of 1 mM TCEP. About four-fold excess of fluorescent dye (tetramethylrhodamine-5-maleimide, TMR) was added to the sample and incubated for about an hour at room temperature with stirring and protected from light. The reaction mixture was then passed through two 5 mL HiTrap Desalting columns (GE Healthcare Biosciences Corp.) equilibrated with 20 mM $KH_2PO_4$, 90 mM KCl, 10 mM $MgCl_2$, 1 mM $NaN_3$ pH 6.5 to remove the unreacted dye. The extent of labeling was also verified using mass spectrometry. For the double-mutant PKIα, the procedure described above was used. In this case, Alexa-488 was added to the solution in 1:1 stoichiometric ratio with PKIα and the reaction was allowed to proceed for one hour at room temperature with stirring and protected from light. Since Alexa Fluor 488 carries a charge, the single-labeled PKIα was separated using anion exchange chromatography with HiTrap Q HP column using an NaCl gradient in 20 mM TRIS pH 7.0. The fraction containing the single Alexa-labeled PKIα was buffer exchanged using a PD-10 MidiTrap column equilibrated with 20 mM $KH_2PO_4$, 90 mM KCl, 10 mM $MgCl_2$, 1 mM $NaN_3$, pH 7.0. Five-fold excess of tetramethylrhodamine-5-maleimide (TMR) was added and the reaction mixture was incubated for about an hour with stirring and protected from light. The asymmetrically double-labeled PKIα was separated from the unreacted dye by passing the reaction through two 5 mL HiTrap Desalting columns (GE Healthcare Biosciences Corp.) equilibrated with 20 mM $KH_2PO_4$, 90 mM KCl, 10 mM $MgCl_2$, 1 mM $NaN_3$ pH 6.5. The extent of labeling in each step was determined using mass spectrometry. For the labeling of PKA-C$^{C199A, S325C}$ mutant, the protein was buffer-exchanged by passing through a PD-10 MidiTrap column (GE Healthcare Bio-Sciences Corp.) equilibrated with 20 mM $KH_2PO_4$, 90 mM KCl, 10 mM $MgCl_2$, 1 mM $NaN_3$, pH 7.0 at room temperature. The fractions containing PKA-C were pooled and the concentration was adjusted to ~5–6 μM. ATP at a final concentration of 8 mM was added to protect the C343 site from being labeled as shown previously (*Nelson and Taylor, 1981*). The reaction was initiated by adding a five-fold molar excess of Alexa Fluor 488 and allowed to proceed for about one hour at room temperature with stirring and protected from light. The sample was then buffer-exchanged with the same buffer composition as before, but with 10 mM DTT and pH 6.5. The extent of labeling was assessed using mass spectrometry. The protein sample was then concentrated to ~20 μM.

## NMR experiments

Backbone resonance assignment of PKIα bound to PKA-C was achieved using standard triple-resonance 3D NMR experiments (*Ikura et al., 1990*; *Kay et al., 2011*). For the assignment of the free form, all the NMR experiments were acquired on a Varian Inova 600 MHz spectrometer equipped with an HCN Cold Probe and on a Bruker 700-MHz Avance spectrometer equipped with a 5 mm triple resonance cryoprobe. The $^{1}$H-$^{15}$N HSQC (*Bodenhausen and Ruben, 1980*) experiments were acquired with 16 scans, 2048 ($^{1}$H) and 100 ($^{15}$N) complex points, before and after each triple-resonance experiment. The HNCACB and CBCA(CO)NH (*Grzesiek and Bax, 1992*) experiments were collected with 64 scans, 1643 ($^{1}$H), 108 ($^{15}$N), and 128 ($^{13}$C) complex points. The HNCO (*Kay et al., 2011*; *Yang and Kay, 1999*) experiments were acquired with 32 scans, 1024 ($^{1}$H), 40 ($^{15}$N), and 54 ($^{13}$C) complex points. Standard $^{15}$N-edited TOCSY-HSQC and NOESY-HSQC (*Marion et al., 1989*) experiments were acquired with 64 scans, 1024 ($^{1}$H) and 120 ($^{15}$N) complex points. The CC(CO)NH-TOCSY (*Cavanagh et al., 2011*) experiment was recorded with 1024 ($^{1}$H), 30 ($^{15}$N), and 70 ($^{13}$C)

complex points, with 32 scans on a Bruker 700 MHz Advance III spectrometer equipped with a 1.7 mm TCI MicroCryoProbe. All data was processed using NMRPipe (*Delaglio et al., 1995*) and visualized using NMRFAM-Sparky (*Goddard and Kneller, 2004*; *Lee et al., 2015*).

All experiments for the backbone assignment of the PKA-C/ATPγN/PKIα ternary complex were performed on a Bruker Avance III 850 MHz spectrometer with a TCI cryoprobe. TROSY-based (*Salzmann et al., 1998*; *Salzmann et al., 1999*) HNCA and HN(CO)CA (*Kay et al., 2011*) experiments were collected with a minimum of 32 scans, 1024 ($^1$H), 32 ($^{15}$N), and 64 ($^{13}$C) complex points. The TROSY-based HNCACB experiment was collected with 32 scans, 1024 ($^1$H), 35 ($^{15}$N) and 50 ($^{13}$C) complex points were performed to measure the $^{13}$Cα and $^{13}$Cβ correlations. The HNCO experiments were acquired with a minimum of 16 scans, 1024 ($^1$H), 30 ($^{15}$N), and 40 ($^{13}$C) complex points. Before and after each triple resonance experiments a $^1$H-$^{15}$N CLEAN-TROSY-HSQC (*Schulte-Herbrüggen and Sorensen, 2000*) spectrum was acquired with 1024 ($^1$H) and a minimum of 64 ($^{15}$N) complex points. All data were processed using NMRPipe (*Delaglio et al., 1995*) and visualized using Sparky (*Lee et al., 2015*; *Goddard and Kneller, 2004*). The CSI for Cα, Cβ, C' and Hα were derived with respect to the reference calculated by Schwarzinger et al. (*Schwarzinger et al., 2000*; *Schwarzinger et al., 2001*) and plotted using the GraphPad Prism six software package (GraphPad Software, Inc). The CSP plot (*Williamson, 2013*) was calculated using the amide chemical shifts according to the following equation:

$$\Delta\delta = \sqrt{\Delta\delta_{HN}^2 + (0.154\Delta\delta_N)^2} \tag{1}$$

Where $\Delta\delta$ is the compounded chemical shift perturbation, $\Delta\delta_{HN}$ is the chemical shift perturbation of the amide proton, $\Delta\delta_N$ is the chemical shift perturbation of nitrogen, and 0.154 is the scaling factor for nitrogen (*Mulder et al., 1999*).

## NMR relaxation experiments

The heteronuclear [$^1$H,$^{15}$N]-NOE spectra were acquired using standard pulse sequences (*Farrow et al., 1994a*) on a Bruker 850 MHz and 900 MHz Advance spectrometer III equipped with TCI cryoprobes at 27°C. The heteronuclear [$^1$H,$^{15}$N]-NOE values were calculated from the ratio of the peak intensities with and without proton saturation. The errors were estimated by evaluating the standard deviation of the NOE values ($\sigma_{NOE}$):

$$\frac{\sigma_{NOE}}{NOE} \sqrt{\left(\frac{\sigma_{I_{sat}}}{I_{sat}}\right)^2 + \left(\frac{\sigma_{I_{unsat}}}{I_{unsat}}\right)^2} \tag{2}$$

where $\sigma_{I_{sat}}$ and $\sigma_{I_{unsat}}$ are the root-mean-square noise of the spectra and $I_{sat}$ and $I_{unsat}$ are the intensities of the resonance with and without proton saturation (*Farrow et al., 1994b*).

The $^{15}$N spin relaxation rates ($T_1$ and $T_2$) for PKIα free and bound to PKA-C/ATPγN complex were measured as reported by *Barbato et al. (1992)* by acquiring a series of 2D [$^1$H,$^{15}$N]-HQCS spectra at different relaxation delays. The relaxation rates for each resided were obtained by fitting the intensities of each peaks with an exponential decay function. 16 scans, 1024 ($^1$H), 60 ($^{15}$N) complex points were used to record $T_1$-relaxation experiment. The $T_2$ measurements were recorded using 32 scans, 1024 ($^1$H), 64 ($^{15}$N) complex points for both PKIα free and bound. For $T_1$, the relaxation delays were: (20), 100, 200, 300, 400, 600, 800, and 1000 ms. For $T_2$, the relaxation delays were (16.6), 83.1, 166, 249, (332), 498, 664, 830 ms. The numbers in parenthesis indicate the experiments that were repeated for error estimation. All the experiments were recorded using a Bruker 900 MHz Advance spectrometer III equipped with TCI cryoprobes at 27°C. All experiments on the free form of PKIα was performed using a 0.2 mM sample of uniformly $^2$H/$^{15}$N labeled protein. To measure PKIα relaxation in the ternary complex, the final sample concentration was 0.180 mM $^2$H/$^{15}$N labeled PKIα and 0.220 mM of U-$^2$H labeled PKA-C. All samples were prepared in aqueous buffer consisting of 20 mM KH$_2$PO$_4$, 90 mM KCl, 60 mM MgCl$_2$, 10 mM DTT, and 1 mM NaN$_3$ at pH 6.5.

The µs-ms timescale conformational dynamics of backbone amides in PKIα free form and the PKA-C/ATPγN/PKIα complex were measured using the relaxation-compensated version of the Carr-Purcell-Meiboom Gill (CPMG) relaxation dispersion experiments (*Baldwin and Kay, 2009*; *Long et al., 2008*; *Palmer et al., 2001*). The experiments were performed in an interleaved mode with a ν$_{CPMG}$ values of 0, 12.5 and 1000 Hz. Effective transverse relaxation rate constant, $R_{2, eff}$, were

determined at each $\nu_{CPMG}$ value using the peak intensities with and without relaxation period according to the equation (*Mulder et al., 2001*):

$$R_{2,\,eff} = \frac{\ln\left(\frac{I_0}{I}\right)}{T} \tag{3}$$

where $I_0$ and $I$ are the peak intensities of the resonances in the 2D spectra acquired in the absence and presence of a relaxation period, respectively, and $T$ is the total CPMG time (*Mulder et al., 2001*) which was 40 ms. All the experiments were acquired on Bruker 900 MHz spectrometer using 16 scans with 1024 ($^1$H) and 64 ($^{15}$N) complex points with a recycle delay of 4.5 seconds.

All the spectra were processed using NMRPipe (*Delaglio et al., 1995*) and visualized using NMRFAM-Sparky (*Goddard and Kneller, 2004*; *Lee et al., 2015*). (T.D. Goddard and D.G. Kneller, UCSF).

## Paramagnetic relaxation measurements

The intra-molecular $^1$H PRE-$\Gamma_2$ relaxation measurements on the free form of PKIα mutants were carried out using the pulse sequence by *Iwahara et al. (2007)* on a Bruker Advance 700 MHz spectrometer at 27°C. All experiments were performed using 160 scans with 2048 ($^1$H) and 128 ($^{15}$N) complex points. A two-time point measurement was performed using a relaxation time of 4 and 14 ms in an interleaved fashion. The $^1$H PRE-$\Gamma_2$ values were calculated using the equation (*Iwahara et al., 2007*):

$$\Gamma_2 = \frac{1}{T_b - T_a} \ln \frac{I_{dia}(T_b)I_{para}(T_a)}{I_{dia}(T_a)I_{para}(T_b)} \tag{4}$$

where $\Gamma 2$ is the PRE-relaxation rate, the time points are $T_a$ and $T_b$, $I_{para}$ is the corresponding intensity with a spin label and $I_{dia}$ is the corresponding intensity with a reduced spin label:

$$\sigma(\Gamma_2) = \frac{1}{T_b - T_a}\sqrt{\left\{\frac{\sigma_{dia}(T_a)}{I_{dia}(T_a)}\right\}^2 + \left\{\frac{\sigma_{dia}(T_b)}{I_{dia}(T_b)}\right\}^2 + \left\{\frac{\sigma_{para}(T_a)}{I_{para}(T_a)}\right\}^2 + \left\{\frac{\sigma_{para}(T_b)}{I_{para}(T_b)}\right\}^2} \tag{5}$$

Where $\sigma_{dia}$ and $\sigma_{para}$ are the roots mean square noise of the respective spectra. For the simultaneous detection of inter- and intra-molecular PREs of PKIα bound to PKA-C/ATPγN, we used a TROSY-based CCLS variation of the standard PRE pulse sequence (*Olivieri et al., 2018*). For both paramagnetic and diamagnetic experiment, the complex sample was prepared with a total protein concentration of 140 μM using a 1:1 PKA-C/PKIα molar ratio.

## Residual dipolar coupling measurements

Backbone amide $^1$D$_{NH}$ RDCs of $^{15}$N PKIα were measured at 27°C by taking the difference in $^1$J$_{NH}$ scalar coupling in aligned and isotropic media (*Tjandra and Bax, 1997*). The aligned medium was constituted by DMPC/D7PC bicelles (q = 3.5) and $^1$J$_{NH}$ couplings were measured using the *in*-phase *anti*-phase [$^1$H-$^{15}$N] HSQC sequence (*Ottiger et al., 1998*). To get an orthogonal set of $^1$D$_{NH}$ RDCs values, we used stretched polyacrylamide gel (*Ishii et al., 2001*). All the experiments were carried out on a Bruker 850 MHz Avance III spectrometer equipped with a 5 mm triple resonance cryoprobe, using 320 scans with 1024 ($^1$H dimension) and 60 ($^{15}$N dimension) complex points.

## Circular dichroism (CD) spectra acquisition and data analysis

The CD spectra in the far-UV of free PKIα in native conditions were recorded using a JASCO J-815 spectropolarimeter (*Chemes et al., 2012*). The spectra were acquired at 25°C scanning from 180 to 260 nm and recording at 20 nm/min scan speed with a bandwidth of 5 nm and a pitch of 0.1 nm. 100 μM PKIα was solubilized in 20 mM PIPES buffer (pH 7.0) with 150 mM NaCl. To record the baseline, a blank sample with buffer alone was used. The data were analyzed using the BeSTSel web server (http://bestsel.elte.hu/; *Micsonai et al., 2018*).

## Small angle X-ray scattering (SAXS) experiment and data analysis

SAXS data were collected at the University of Utah using an Anton Paar SAXess instrument with line collimation and solid-state detector. In both PKIα free and the PKA-C/PKIα complex, the proteins were equilibrated by dialysis in buffer containing 20 mM MOPS, 90 mM KCl, 60 mM MgCl$_2$, 10 mM

DTT, 1 mM NaN$_3$ (pH 6.5), and 12 mM of ATPγN. For measurements of the PKIα free form, a protein concentration of 1 mM was used, and for the PKA-C/PKIα complex sample, a final protein concentration 0.2 mM protein with PKA-C/PKIα in 1:1 molar ratio. SAXS data were reduced and desmeared using the SAXSQuant (Anton Paar) software suite and further analyzed using PRIMUS in the ATSAS 2.8.3 software suite (*Franke et al., 2017*). The fitting of the MD structural ensembles to the experimental scattering data was performed using the MultiFoXS server (*Schneidman-Duhovny et al., 2013*). 100 random snapshots from MD simulations with different radius of gyration ($R_g$) were chosen as the conformational ensembles, and the fitting was done for the $q$ range (0.01 to approximately 0.18 Å$^{-1}$). The top 10 structures with the lowest χ values were selected as the final structure ensembles.

## Modeling of PKIα ensembles using replica-averaged metadynamics (RAM) simulations

For the free form, we used the extended conformation of the full-length PKIα as a template. The protein was solvated in a rhombic dodecahedron solvent box with TIP3P water molecule layer extended approximately 10 Å away from the protein's surface. Counter ions (K$^+$ and Cl$^-$) were added to ensure electrostatic neutrality corresponding to an ionic concentration of ~150 mM. All protein covalent bonds were constrained with the LINCS (*Hess et al., 1997*) algorithm and long-range electrostatic interactions were calculated using the particle-mesh Ewald method with a real-space cutoff of 10 Å (*Best et al., 2012*; *Darden et al., 1993*). Parallel simulations of 130 ns were performed simultaneously using GROMACS 4.6 (*Hess et al., 2008*) with CHARMM36a1 force field (*Best et al., 2012*). As a starting configuration for the bound form, we used the ternary complex of PKA-C/ATP/PKI$_{5-24}$ (PDB: 1ATP) and built the reminder amino acids of PKIα using PyMOL (pymol.com). The remaining parameters were set using the same protocol for the free form.

The subsequent RAM simulations were started from the random snapshots of the production trajectory (the last 100 ns). As for previous RAM simulations (*Roux and Weare, 2013*), we chose 10 replicas: five replicas initiated from partially folded conformers, and the others from extended conformers. We set two protocols imposing different restraints: a) RAM with CS restraints (RAM_CS) for both free and bound forms in which the predicted backbone chemical shifts were back-calculated by Camshift method implemented in ALMOST 2.1 (*Fu et al., 2014*); and b) RAM with CS and RDC restraints (RAM_CSRDC) for the free form only in which the predicted RDCs were back-calculated using the tensor-free method (*Camilloni and Vendruscolo, 2015*) implemented in PLUMED 2.1 (*Bonomi et al., 2009*). Four collective variables (CVs) were chosen to monitor the conformational plasticity and secondary structure of PKIα: a) psi angles for all residues (backbone and dihedral); b) radius of gyration for all backbone Cα atoms (radius of gyration); c) helical content of residues 4–22 (helical content of the N-terminus); and d) helical content of residue 32–50 (helical content of the NES motif). Gaussian deposition rate was performed with an initial rate of 0.125 kJ/mol/ps, where the σ values were set to 0.5, 0.02 nm, 0.1, and 0.1 for the four CVs. Of the 10 replicas, each of the four CVs were imposed on two replicas, and the remaining two replicas were left neutral. The RAM simulations were carried out using GROMACS 4.6 in conjunction with PLUMED 2.1 (*Bonomi et al., 2009*) and ALMOST 2.1 (*Fu et al., 2014*), and continued for ~250 ns for each replica with exchange trails every 0.5 ps. The free energy along the four CVs converged after 150 ns, with the standard deviation below one kcal/mol. The last 100 ns of each replica constituted the production trajectory for a total simulation time of 1.0 μs and used for analysis.

## Adaptive sampling of pkiα in free and bound forms and Markov model analysis

For the free form, the adaptive sampling was started from randomly chosen 320 snapshots of PKIα from the RAM simulations. The initial velocities were randomly generated to satisfy the Maxwell distribution at 300K. A ~ 200 ns simulation was initiated from each starting configuration. Therefore, a total of 60 μs trajectories and 240,000 snapshots (250 ps per frame) were collected for subsequent analysis. For the bound form, the adaptive sampling was started from 50 snapshots of the RAM simulations, and ~20 μs trajectories have been collected. More samplings are undertaken.

## Markov model and time-lagged independent component analyses (tICA)

The Markov model analysis was carried out using MSMBuilder 3.5 (*Harrigan et al., 2017*). The contact distances between residue pairs that are separated by at least four residues, were chosen as the metrics to characterize the conformational transition of PKIα. The representation in this metric space was further reduced to 6-dimension vectors using time-lagged independent component analysis (tICA) at a lag time of 15 ns. The 240000 snapshots were clustered into 50 microstates with mini-batch K-Mean clustering and Markov model was built upon the transition counts between these microstates. The same tICA analysis was applied to both free and bound forms.

## Statistical analysis of equilibrium population and MFPT

To obtain the statistical distribution and errors on the equilibrium population and MFPT along the given tICA projections, 150 rounds of bootstrapping with replacement were performed. Specifically, for each round, new clustering and Markov model is built, and the microstates are categorized into four major states based on the tICA projections. The equilibrium population and MFPT are subsequently computed. The same MFPT analysis was applied to both free and bound forms.

## Stopped-flow TR-FRET experiments and kinetic binding model

The kinetics of binding between PKA-C and PKIα were measured using home-built stopped-flow spectrofluorometer (*Muretta et al., 2015*). Equal volumes of each component were mixed, the samples were then excited at 475 nm, and fluorescence emission was measured using a 520 nm filter. For Alexa-labeled PKA-C$^{C199A, S325C}$ and TMR-labeled PKIα, the final concentration of the kinase was fixed at 100 nM, while the concentration of PKIα was varied from 0 to 250 times the concentration of PKA-C. For the doubly-labeled PKIα$^{V3C, S59C}$ and unlabeled PKA-C, the concentration of the peptide was fixed at 100 nM final concentration while the concentration of the kinase was varied from 0 to 800 times the concentration of PKIα$^{V3C, S59C}$. The buffer used in both setups contains 20 mM KH$_2$PO$_4$, 90 mM KCl, 10 mM MgCl$_2$, 16 mM ATP, 10 mM DTT, 1 mM NaN$_3$, pH 6.5 at 25°C. The total fluorescence was fit into equations describing either single, double or triple exponential decays. Data analyses were done using Origin (OriginLab) software.

Since we saturated PKA-C with an excess of nucleotide, we neglected the equilibrium between open (apo PKA-C) and intermediate (ATP-bound, PKA-C*) state in the fitting procedure. In the model, PKI* represents the ensemble of conformations that is recognized by PKA-C*. The proposed model of binding is reported in *Scheme 1*.

The differential equations describing the kinetic model of *Scheme 1* are:

$$\frac{d[P^*]}{dt} = +k_{-b}[P^*I^*] - k_b[P^*][I^*] \tag{6}$$

$$\frac{d[I]}{dt} = k_{-1}[I^*] - k_1[I] \tag{7}$$

$$\frac{d[I^*]}{dt} = k_1[I] - k_{-1}[I^*] + k_{-b}[P^*I^*] - k_b[P^*][I^*] \tag{8}$$

$$\frac{d[P^*I^*]}{dt} = +k_b[P^*][I^*] - k_{-b}[P^*I^*] - k_r[P^*I^*] + k_{-r}[(PI)^*] \tag{9}$$

$$\frac{d[(PI)^*]}{dt} = +k_r[P^*I^*] - k_{-r}[(PI)^*] \tag{10}$$

For simplicity, PKA-C is denoted as P and PKI as I. The instantaneous fluorescence intensity FI was calculated using the formula,

$$FI = \frac{(c[PKA\ C^*\ PKI^*] + c^*[\ (PKA\ C\ \ PKI)^*])}{(c[PKA\ C^*\ PKI^*]eq + c^*[\ (PKA\ C\ \ PKI)^*]eq)} \times SC + Baseline \tag{11}$$

Where $\frac{(c[PKA\ C^*\ PKI^*] + c^*[\ (PKA\ C\ \ PKI)^*])}{(c[PKA\ C^*\ PKI^*]eq + c^*[\ (PKA\ C\ \ PKI)^*]eq)}$ is the normalized intensity, 'SC' is a scaling factor, 'Baseline'

is the basal fluorescence, c and c* are the FRET efficiency parameters. The differential *equations 1-5* were solved numerically using MATLAB ode solver 'ode23s', which was incorporated in an optimization protocol to evaluate the kinetic parameters. The optimization protocol included an initial minimization using MATLAB's nonlinear programming solver (fminsearch), which was ran for $10^4$ steps and followed by Genetic Algorithm optimization with a population size of 50 and $10^4$ generations. All the six rate constants ($k_1$, $k_{-1}$, $k_b$, $k_{-b}$, $k_r$, $k_{-r}$) were collectively fit for all data points. The FRET efficiency parameters (c and c*) were fit for each specific labelling scheme. The parameters 'SC' and 'Baseline' were fit for each FRET experiment to correct for the experimental errors.

The fitting resulted in an on rate binding constant ($k_b$) of $1.5 \times 10^7$ M$^{-1}$s$^{-1}$ and a dissociation rate constant ($k_{-b}$) of 0.5 s$^{-1}$. For the slow phase, we found a forward rate of 2400 s$^{-1}$ and reverse rate of 720 s$^{-1}$. The 'on' and 'off' conformational exchange rates ($k_1$ and $k_{-1}$) of free PKIα were $1.9 \times 10^4$ s$^{-1}$ and $1.5 \times 10^4$ s$^{-1}$.

## Acknowledgements

This work was supported by the National Institute of Health GM 100310 to GV and GM46736 to JG. The authors would like to acknowledge Dr. FW Herberg for sharing the plasmid of PKIα as well as the Minnesota Supercomputing Institute for MD calculations. The SAXS instrument at the University of Utah was funded by the Department of Energy (Dr. J Trewhella).

## Additional information

### Funding

| Funder | Grant reference number | Author |
| --- | --- | --- |
| National Institute of General Medical Sciences | GM 100310 | Gianluigi Veglia |
| National Institute of General Medical Sciences | GM 46736 | Jiali Gao |

The funders had no role in study design, data collection and interpretation, or the decision to submit the work for publication.

### Author contributions

Cristina Olivieri, Data curation, Formal analysis, Methodology, Writing - review and editing; Yingjie Wang, Geoffrey C Li, Formal analysis, Methodology, Writing - original draft, Writing - review and editing; Manu V S, Data curation, Software, Formal analysis, Visualization, Writing - original draft, Writing - review and editing; Jonggul Kim, Formal analysis, Visualization, Writing - review and editing; Benjamin R Stultz, Matthew Neibergall, Methodology, Writing - review and editing; Fernando Porcelli, Formal analysis, Writing - original draft, Writing - review and editing; Joseph M Muretta, Data curation, Formal analysis, Methodology, Writing - original draft, Writing - review and editing; David DT Thomas, Conceptualization, Supervision, Writing - original draft, Writing - review and editing; Jiali Gao, Conceptualization, Supervision, Validation, Methodology, Writing - original draft, Writing - review and editing; Donald K Blumenthal, Conceptualization, Formal analysis, Investigation, Methodology, Writing - review and editing; Susan S Taylor, Conceptualization, Writing - original draft, Writing - review and editing; Gianluigi Veglia, Conceptualization, Data curation, Supervision, Funding acquisition, Validation, Methodology, Writing - original draft, Project administration, Writing - review and editing

### Author ORCIDs

Cristina Olivieri (ID) https://orcid.org/0000-0001-6957-6743
Geoffrey C Li (ID) https://orcid.org/0000-0001-5035-5916
Jonggul Kim (ID) https://orcid.org/0000-0002-4624-7848
Fernando Porcelli (ID) https://orcid.org/0000-0003-3209-0074
David DT Thomas (ID) https://orcid.org/0000-0002-8822-2040

Susan S Taylor ⬦ http://orcid.org/0000-0002-7702-6108
Gianluigi Veglia ⬦ https://orcid.org/0000-0002-2795-6964

**Decision letter and Author response**
Decision letter https://doi.org/10.7554/eLife.55607.sa1
Author response https://doi.org/10.7554/eLife.55607.sa2

## Additional files

### Supplementary files
• Transparent reporting form

### Data availability

All NMR data are deposited as NMR-Star File in the BMRB (BMRB entries 50238, 50243).

The following datasets were generated:

| Author(s) | Year | Dataset title | Dataset URL | Database and Identifier |
|---|---|---|---|---|
| Veglia G, Olivieri C, Li GC, Muretta J | 2020 | Analysis of the kinetics of binding of Protein Kinase A Inhibitor alpha (PKIa) to cAMP-dependent protein kinase a catalytic subunit (PKA-C) | https://conservancy.umn.edu/handle/11299/212383 | University of Minnesota repository, 212383 |
| Olivieri C, VS M, Veglia G | 2020 | Backbone (1H, 13C and 15N) Chemical Shift Assignments and 15N Relaxation Parameters for protein kinase Inhibitor alpha (PKIa) free state | http://www.bmrb.wisc.edu/data_library/summary/index.php?bmrbId=50238 | Biological Magnetic Resonance Data Bank, 50238 |
| Olivieri C, VS M, Veglia G | 2020 | Backbone (1H, 13C and 15N) Chemical Shift Assignments and 15N Relaxation Parameters for protein kinase Inhibitor alpha (PKIa) bound to cAMP-dependent protein kinase A | http://www.bmrb.wisc.edu/data_library/summary/index.php?bmrbId=50243 | Biological Magnetic Resonance Data Bank, 50243 |

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
