## [Decision Letter]

Thank you for submitting your article "Multi-state recognition pathway of the intrinsically disordered protein kinase inhibitor by protein kinase A" for consideration by *eLife*. Your article has been reviewed by two peer reviewers, and the evaluation has been overseen by a Reviewing Editor and José Faraldo-Gómez as the Senior Editor.

The reviewers have discussed the reviews with one another and the Reviewing Editor has drafted this decision to help you prepare a revised submission.

Summary:

Thank you for submitting this excellent work for consideration at *eLife*. The reviewers were effusive in their praise of the work with some of the comments indicated below:

– "The article by Olivieri et al. reports an exceptional analysis of the mechanism of binding and biological activation of a key complex by the cAMP-dependent protein and its inhibitor PKI to recruit CRM1/RanGTP…"

– "The work analyzes at high resolution the structural, kinetic and mechanistic properties of the interaction between the kinase and its inhibitor to achieve a conclusive biochemical model…"

– "This is extremely well performed research that spans a large set of biophysical experiments and computations to obtain a cutting-edge characterisation of an important biological mechanism. The work is also very well written."

– "Overall, I find this manuscript suitable for publication here given the benefit of extending long-standing PKA/PKI complex information to include additional portions required for function."

As you can see both reviewers feel that this paper will be suitable for publication, but they also agree that the work would benefit from additional details concerning the methods. It is also suggested that you try to underscore the functional importance more clearly at the outset of the manuscript – aspects of it seemed better described elsewhere, but the motivation for the work (and in turn, the gaps in knowledge that it addresses) would be better emphasized early on. Importantly, no further experiments are required.

Essential revisions:

1) NMR analysis of the complex

The study of the complex PKA-C/ATPgN/PKIα employed a large set of NMR experiments, performed with the highest standards and using a careful choice of the experimental conditions and analyses. The data from ^15^N relaxation (T1, T2, and hetNOE), CPMG, CSP and PREs show conclusively that PKI has a structural rearrangement upon binding, with the HAR and PSS motifs experiencing a disorder-to-order transition. Additional conformational transitions were also found in various regions of PKA upon binding.

The presentation of the results is very detailed, however, the readability for a non NMR-expert audience can be improved. For example, in subsection “PKA-C binding rigidifies the PSS motif and stabilizes HAR and NES helices of PKIα”:

– "Also, using the [^1^H,^15^N]-CCLS-HSQC experiment…we were able to map…"; please explain how the CCLS enables to map the fingerprints of PKA-C and PKI in the complex simultaneously.

– "where both the HAR and PSS motifs show an increase in longitudinal relaxation and a concomitant decrease in the transverse relaxation rates"; please explain for a non-expert audience what is the structural and molecular interpretation of these spectral properties.

– The rigidification of the PSS motif and the increase in helical content of the HAR are clear from the CS analysis, however, the wider audience (non-NMR-experts) may not fully appreciate this from the plots reported in Figures 2A and 2B. One possibility may be to present the results in a different manner, e.g. using RCI (Wishart) to provide a more direct representation of the dynamic content of the protein via an elaboration of the CS data, or delta2D (Vendruscolo) to directly show the content of secondary structure from CS. This reviewer, however, appreciate the approach of the authors in showing raw data rather than fitted data from the NMR observables.

– "suggesting that the peptide does not experience". Is this a typo with the correct sentence "suggesting that the peptide does experience"?

2) MD restraints

Replica averaged metadynamics simulations were run on the free and kinase bound PKIα. These simulations are not standard, although have now become quite used in the community, and therefore require a very detailed level of description in the Materials and methods section.

For example, the ALMOST 2.1 algorithm is cited for CS and RDC restraints, however, the associated reference (Kohlhoff et al., 2009) refers to an evolution of ALMOST, namely camshift, which is a method to predict chemical shifts from interatomic distances that can be used as a restrain in MD simulations. Moreover, the Kohlhoff et al. reference does not cover the RDCs, which were also used in the present restrained simulations. It is important to describe the way by which RDCs are modelled (e.g. SVD vs ab initio prediction of the alignment tensor) as this can make a significant difference in the effect of RDC restraints in the resulting concerted motions of proteins (see JCTC 2011 7:4189-419).

Additional elements of description:

– the methods report that "CS restraints were imposed as averaged over the 10 replicas" but do not say how were RDC restrained?

– Can the authors justify the choice of 10 replicas? Number of replicas is an essential choice in ensemble averaged NMR restraints to avoid over-fitting or over-restraining.

3) Convergence of the simulations

The quality of RAM simulations (as essentially in all biomolecular simulations) is crucially dependent on the convergence, however, this aspect doesn't seem to be described in the manuscript. Could the authors add a description of the convergence criteria employed?

Introduction, third paragraph: "More importantly, the mechanism of PKI recognition and binding is essentially unknown." Seemingly at odds with existence of many structures and corresponding biochemistry, please be more specific.

Subsection “PKIα is an intrinsically disordered protein with transient secondary and short-lived tertiary interactions”/Figure S2B-D: it is unclear how residues were determined to have a "strong" or "marked" PRE, please comment further. Also clarify the meaning of yellow dots in Figure S2B-D, which presumably indicate residues with no signal detected.

Figure 2C-E: what do colored horizontal lines represent in Figure 2C, D? Presumably something with respect to significance, as in Figure 2A, but not stated. Also, not entirely clear where the complex structure shown in Figure 2E (and again, in several figures afterwards) comes from – although a comment in Figure 4 legend suggests that this is a model starting from Knighton et al., 1991a. Please cite, and comment explicitly about the fact that a single conformation is shown for the C-term of PKI (which is later shown to be quite flexible).

Figure 4C and commentary in text ("PKIα undergoes large amplitude motions, featuring an equilibrium between elongated and more folded conformations that transiently interact…") is not really well linked to presented data, please clarify. Assuming that this implies that PKIα elongated in apo- form – moderately well shown by PRE, but is this rigorous enough to rule out partially-collapsed state? – and kinase-bound in different conformation. Seems better justified after FRET data shown subsequently. Also please explicitly mark PKI position 59 in Figure 4C to highlight MTSL labeling location.

Figure 6—figure supplement 1: list concentration range depicted by different colored lines, please.

---

## [Author Response]

Essential revisions:1) NMR analysis of the complexThe study of the complex PKA-C/ATPgN/PKIα employed a large set of NMR experiments, performed with the highest standards and using a careful choice of the experimental conditions and analyses. The data from 15N relaxation (T1, T2, and hetNOE), CPMG, CSP and PREs show conclusively that PKI has a structural rearrangement upon binding, with the HAR and PSS motifs experiencing a disorder-to-order transition. Additional conformational transitions were also found in various regions of PKA upon binding.The presentation of the results is very detailed, however, the readability for a non NMR-expert audience can be improved. For example, in subsection “PKA-C binding rigidifies the PSS motif and stabilizes HAR and NES helices of PKIα”:– "Also, using the [1H,15N]-CCLS-HSQC experiment…we were able to map…"; please explain how the CCLS enables to map the fingerprints of PKA-C and PKI in the complex simultaneously.

We thank this reviewer for the suggestion. We addressed this point in the revised version and we have also changed the manuscript, making it more accessible to non-NMR experts.

– "where both the HAR and PSS motifs show an increase in longitudinal relaxation and a concomitant decrease in the transverse relaxation rates"; please explain for a non-expert audience what is the structural and molecular interpretation of these spectral properties.

Thank you for pointing this out. Again, we revised the text and eliminated unnecessary jargon, making it more accessible to non-NMR experts.

– The rigidification of the PSS motif and the increase in helical content of the HAR are clear from the CS analysis, however, the wider audience (non-NMR-experts) may not fully appreciate this from the plots reported in Figures 2A and 2B. One possibility may be to present the results in a different manner, e.g. using RCI (Wishart) to provide a more direct representation of the dynamic content of the protein via an elaboration of the CS data, or delta2D (Vendruscolo) to directly show the content of secondary structure from CS. This reviewer, however, appreciate the approach of the authors in showing raw data rather than fitted data from the NMR observables.

As per this reviewer’s request, we included two new figures with the requested plots (see Figure 2B and Figure 3—figure supplement 3A). We also included the references for the above cited authors.

– "suggesting that the peptide does not experience". Is this a typo with the correct sentence "suggesting that the peptide does experience"?

Yes, this was typo and we corrected in the revised version.

2) MD restraintsReplica averaged metadynamics simulations were run on the free and kinase bound PKIα. These simulations are not standard, although have now become quite used in the community, and therefore require a very detailed level of description in the Materials and methods section.For example, the ALMOST 2.1 algorithm is cited for CS and RDC restraints, however, the associated reference (Kohlhoff et al., 2009) refers to an evolution of ALMOST, namely camshift, which is a method to predict chemical shifts from interatomic distances that can be used as a restrain in MD simulations. Moreover, the Kohlhoff et al. reference does not cover the RDCs, which were also used in the present restrained simulations. It is important to describe the way by which RDCs are modelled (e.g. SVD vs ab initio prediction of the alignment tensor) as this can make a significant difference in the effect of RDC restraints in the resulting concerted motions of proteins (see JCTC 2011 7:4189-419).Additional elements of description:– the methods report that "CS restraints were imposed as averaged over the 10 replicas" but do not say how were RDC restrained?

We agree. We did not report these details in our initial submission. Indeed, the RDC restraints were imposed using the tensor-free method implemented in the PLUMED algorithm (Camilloni and Vendruscolo, 2015), which utilized the direct dependence of RDC on the angle between the internuclear vectors and the external magnetic field. More specifically, we chose 70 experimental RDC values measured using stretched gels as an aligning medium and implemented the restraints to minimize the deviation between calculated and experimental angles (θ). We now provide a more detailed protocol in the Materials and method section and cite the above papers.

– Can the authors justify the choice of 10 replicas? Number of replicas is an essential choice in ensemble averaged NMR restraints to avoid over-fitting or over-restraining.

We thank the reviewer for pointing this out. Replica exchange molecular dynamics simulations is statistically equivalent to the maximum entropy principle (Pitera and Chodera, 2012; Cavalli, Camilloni et al., 2013; Roux and Weare, 2013). To this extent, several authors demonstrated that the probability distribution obtained from 10 replicas is essentially similar to that of 100 replicas as they both converge to a similar distribution of conformers (Pitera and Chodera, 2012; Roux and Weare, 2013). For our calculations, we tested ensembles with both 4 and 10 replicas. We found that the 10 replicas gave an excellent agreement with experimental data, which is why we stopped the calculations. We now clarify this in the revised version.

3) Convergence of the simulationsThe quality of RAM simulations (as essentially in all biomolecular simulations) is crucially dependent on the convergence, however, this aspect doesn't seem to be described in the manuscript. Could the authors add a description of the convergence criteria employed?

To determine the convergence of our simulations, we followed the common protocol described for metadynamics simulations by Vendruscolo et al. Specifically, we carefully monitored the change of free energy along the chosen CV. For our system, the free energy converged with the standard deviation less than 1 kcal/mol after 150 ns of simulation. Then, we continued the simulations for another 100 ns for the production run. We have added these details in the Materials and method section.

Introduction, third paragraph: "More importantly, the mechanism of PKI recognition and binding is essentially unknown." Seemingly at odds with existence of many structures and corresponding biochemistry, please be more specific.

This is an important point. There are several crystal structures of PKA-C/PKI complex with short peptides corresponding to the recognition sequence as well as many inhibition and binding assays to determine the thermodynamic constants of binding. To the best of our knowledge, there are no studies devoted to the mechanism of assembly of the PKA-C/PKI complex neither for truncated peptides or full-length PKI. We further clarify this point in the revised text.

Subsection “PKIα is an intrinsically disordered protein with transient secondary and short-lived tertiary interactions”/Figure S2B-D: unclear how residues were determined to have a "strong" or "marked" PRE, please comment further.

We agree. We now indicate the different categories of PRE effects (weak, medium, and strong) based on the deviation from the average effects among the different samples (see also Figure 2E-G (formerly Figure S2B-D) and Figure 5A,B).

Also clarify the meaning of yellow dots in Figure S2B-D, which presumably indicate residues with no signal detected.

We thank this reviewer for pointing this out. Indeed, the yellow dots indicate the resonances that are broaden out beyond detection for the paramagnetic protein. We changed the figure legends to clarify this.

Figure 2C-E: what do colored horizontal lines represent in Figure 2C,D? Presumably something with respect to significance, as in Figure 2A, but not stated.

We uniformed the colors of the horizontal lines, which indicate one standard deviation of the average chemical shift perturbations (CSP). We clarify this in the figure legends.

Also, not entirely clear where the complex structure shown in Figure 2E (and again, in several figures afterwards) comes from – although a comment in Figure 4 legend suggests that this is a model starting from Knighton et al., 1991a. Please cite, and comment explicitly about the fact that a single conformation is shown for the C-term of PKI (which is later shown to be quite flexible).

We agree with the reviewer. The structure of the complex is a selected conformer from the structural ensemble generated with MD simulations. We clarify this point in the Figure 3E legend.

Figure 4C and commentary in text ("PKIα undergoes large amplitude motions, featuring an equilibrium between elongated and more folded conformations that transiently interact…") not really well linked to presented data, please clarify. Assuming that this implies that PKIα elongated in apo- form – moderately well shown by PRE, but is this rigorous enough to rule out partially-collapsed state? – and kinase-bound in different conformation. Seems better justified after FRET data shown subsequently.

We agree with this comment. We removed the sentence. We did not include in the FRET analysis since we already expressed the same concept in subsection “The PKA-C/PKIα complex relaxes through two structurally and kinetically distinct states to form the complex”.

Also please explicitly mark PKI position 59 in Figure 4C to highlight MTSL labeling location.

We modified Figure 5C by adding a yellow sphere in the location of the spin label. We also changed the figure legend accordingly.

Figure 6—figure supplement 1: list concentration range depicted by different colored lines, please.

The color legends as well as the protein concentrations are now included in Figure 6 and Figure 6—figure supplement 1.